# A kinase-deficient *NTRK2* splice variant predominates in glioma and amplifies several oncogenic signaling pathways

Siobhan S. Pattwell [1], Sonali Arora[1], Patrick J. Cimino [1,2], Tatsuya Ozawa[3], Frank Szulzewsky [1], Pia Hoellerbauer[1,4], Tobias Bonifert[1], Benjamin G. Hoffstrom[5], Norman E. Boiani[5], Hamid Bolouri[1,6], Colin E. Correnti[7], Barbara Oldrini[8], John R. Silber[9], Massimo Squatrito [8], Patrick J. Paddison[1,4] & Eric C. Holland [1,9,10]✉

Independent scientific achievements have led to the discovery of aberrant splicing patterns in oncogenesis, while more recent advances have uncovered novel gene fusions involving neurotrophic tyrosine receptor kinases (NTRKs) in gliomas. The exploration of *NTRK* splice variants in normal and neoplastic brain provides an intersection of these two rapidly evolving fields. Tropomyosin receptor kinase B (TrkB), encoded *NTRK2*, is known for critical roles in neuronal survival, differentiation, molecular properties associated with memory, and exhibits intricate splicing patterns and post-translational modifications. Here, we show a role for a truncated *NTRK2* splice variant, TrkB.T1, in human glioma. TrkB.T1 enhances PDGF-driven gliomas in vivo, augments PDGF-induced Akt and STAT3 signaling in vitro, while next generation sequencing broadly implicates TrkB.T1 in the PI3K signaling cascades in a ligand-independent fashion. These TrkB.T1 findings highlight the importance of expanding upon whole gene and gene fusion analyses to include splice variants in basic and translational neuro-oncology research.

[1] Human Biology Division, Fred Hutchinson Cancer Research Center, 1100 Fairview Avenue North, Mailstop C3-168, Seattle, WA 98109, USA. [2] Department of Pathology, University of Washington School of Medicine, 325 9th AvenueBox 359791Seattle, WA 98104, USA. [3] Division of Brain Tumor Translational Research, National Cancer Center Research Institute, 5-1-1 Tsukiji, Chuo-ku, Tokyo 104-0045, Japan. [4] Molecular and Cellular Biology Program, University of Washington, Seattle, WA 98195, USA. [5] Antibody Technology Resource, Fred Hutchinson Cancer Research Center, 1100 Fairview Avenue North, Seattle, WA 98109, USA. [6] Systems Immunology, Benaroya Research Institute at Virginia Mason, 1201 Ninth Avenue, Seattle, WA 98101, USA. [7] Clinical Research Division, Fred Hutchinson Cancer Research Center, 1100 Fairview Avenue North, Seattle, WA 98109, USA. [8] Seve Ballesteros Foundation Brain Tumor Group, Spanish National Cancer Research Centre, 28209 Madrid, Spain. [9] Department of Neurological Surgery, Alvord Brain Tumor Center, University of Washington School of Medicine, Seattle, WA 98104, USA. [10] Seattle Tumor Translational Research Center, Fred Hutchinson Cancer Research Center, 1100 Fairview Avenue North, Seattle, WA 98109, USA. ✉email: eholland@fredhutch.org

As decades of research have uncovered key drivers in oncogenic signaling through the discovery of tumor suppressors, oncogenes, histone modifications, DNA methylation, and environmental factors, the possibility remains that such factors may contribute to splicing events at the core of oncogenesis. While recent work independently highlights the evolving importance of aberrant splicing in cancer[1–3], simultaneous discoveries have implicated novel *NTRK2* fusions in various glioma subtypes[4–10], yet little is known about endogenous *NTRK2* splicing in human brain or its potential role in brain tumor biology. Prior studies have implicated TrkB in the survival of brain tumor initiating cells in the absence of growth factors epidermal growth factor (EGF) and fibroblast growth factor (FGF)[11], while more recent work has implicated TrkB and its ligand, brain-derived neurotrophic factor (BDNF), in the crosstalk between glioma stem cells and their differentiated glioblastoma cell progeny[12], suggesting that this neurotrophin receptor exhibits complex interactions within the brain tumor environment that extend beyond the canonical TrkB-BDNF signaling events characterized in normal neurodevelopment. Malignant tumors of the central nervous system and brain tumors, specifically, result in the highest years of potential life lost compared with other cancer types[13], while glioblastoma multiforme (GBM), in particular, remains the most common malignant primary brain tumor with a mere 2–4% 5-year survival rate[14]. We sought to further understand the complex role of TrkB in GBM and lower grade gliomas (LGGs) in effort to learn more about the neurotrophin receptor splicing contributions to these devastating tumors.

The neurotrophin receptor TrkB, encoded by the *NTRK2* gene (hg19: chr9:87,283,466-87,638,505) has well known roles in neuronal survival, proliferation, differentiation, apoptosis, and exerts diverse effects on cellular and neural outcomes[15]. In addition to the full-length receptor tyrosine kinase, TrkB.FL, several lesser known, alternatively spliced variants, including the truncated isoform, TrkB.T1, have been shown to exist[16,17]. Once thought dominant-negative due the absence of a kinase domain, TrkB.T1 shares the same extracellular and transmembrane domains, as well as the first 12 intracellular amino acids, as other variants yet contains a unique C-terminal sequence of 11 amino acids that is conserved across species from rodents to humans[17,18]. In vitro, TrkB.T1 has been shown to alter $Ca^{2+}$ signaling[19], regulate neuronal complexity[20,21], influence astrocytic morphology via Rho GTPases[22], modify filopodia outgrowth[23], and contribute to signal transduction and proliferation[22,24,25], raising the possibility that this formerly considered dominant-negative receptor variant has unique and important roles in both normal and abnormal brain development.

Here, we show that the TrkB.T1 splice variant is the predominant TrkB isoform expressed across a range of human gliomas. By generating an antibody specific for this splice variant, we show that TrkB.T1 receptor localization differs between normal, healthy brain regions and gliomas, in both rodents and humans. In vivo experiments using RCAS-tv/a technology demonstrate that TrkB.T1 enhances PDGFB-driven tumors in mice, while in vitro experiments show that TrkB.T1 enhances the perdurance of PI3K and STAT3 signaling pathways including pAkt and pS6rp. Together, these results demonstrate a previously unidentified role for the *NTRK2* splice variant TrkB.T1 in gliomas and highlight the importance of exploring alternative splicing of TRKs in basic and translational research.

## Results

### Distinct gene expression in normal human brain vs. glioma.
To first investigate the overall genetic variance in human brain

**Table 1 GTEx normal brain regions and TCGA tumor types.**

|  | Number of samples |
| --- | --- |
| **GTEx region** |  |
| Normal brain | 1216 |
| Pooled normal brain (spinal cord and cerebellum removed) | 926 |
| Amygdala | 72 |
| Anterior cingulate cortex (BA24) | 82 |
| Caudate (basal ganglia) | 113 |
| Cerebellar hemisphere | 101 |
| Cerebellum | 121 |
| Cortex | 109 |
| Frontal cortex (BA9) | 104 |
| Hippocampus | 92 |
| Hypothalamus | 90 |
| Nucleus accumbens (basal ganglia) | 109 |
| Putamen (basal ganglia) | 93 |
| Spinal cord (cervical c-1) | 68 |
| Substantia nigra | 62 |
| **TCGA tumor type** |  |
| Low grade glioma | 532 |
| Low grade glioma (CIMP) | 437 |
| Low grade glioma (nonCIMP) | 95 |
| GBM | 170 |
| GBM (CIMP) | 5 |
| GBM (nonCIMP) | 59 |
| GBM CIMP status not-available | 106 |

tumors compared with normal brain, we queried publicly available gene expression data from The Genotype-Tissue Expression (GTEx) Project on 1216 normal samples across 13 GTEx-defined brain regions (Table 1), 170 GBM samples and 532 LGG samples from The Cancer Genome Atlas (TCGA). Principal component analysis (PCA) of gene expression data from 20,214 genes across 1216 normal brain GTEx samples revealed three genetically distinct clusters within normal brain samples consisting of multiple supratentorial regions, cerebellum, and spinal cord (Fig. 1a). For a more anatomically accurate comparison of genetic variance from normal brain to brain tumors, cerebellum and spinal cord were removed, supratentorial regions were pooled, and these 926 samples are referred to as Normal Brain (Fig. 1b). PCA of the 20,214 genes Normal Brain, LGG and GBM samples, revealed distinct differences in overall gene expression between normal brain and cancer samples (Fig. 1c). Similar to previous studies showing distinct clustering based on methylation phenotype[26], PCA of gene expression data from GBM and LGG samples also yielded distinct clusters of gliomas based on status of CpG island methylator phenotype (CIMP) and non-CIMP (Table 1; Fig. 1d). To rule out systematic differences between GTEx and TCGA data, we compared median whole gene expression in Normal Brain, LGG and GBM samples at a per-gene level, and found that 10,910 genes were overrepresented in TCGA (7769 in GBM; 3141 in LGG) and 9304 genes had higher expression value in Normal Brain samples. (Supplementary Data 1). After also confirming that *NTRK2* RNASeq expression is reliable across bioinformatic pipelines and does not fall into the less than 12% of genes that are discordantly quantified across commonly used pipelines across[27], we sought compare *NTRK2* expression across normal brain regions and brain tumors. Given the abundance of literature highlighting the role of *NTRK2* in cancer, we found similar whole gene *NTRK2* expression across Normal Brain, LGG and GBM samples, suggesting that differences in expression of particular splice variants may underlie potential oncogenic effects driven by *NTRK2* (Table 2, Supplementary Data 1).

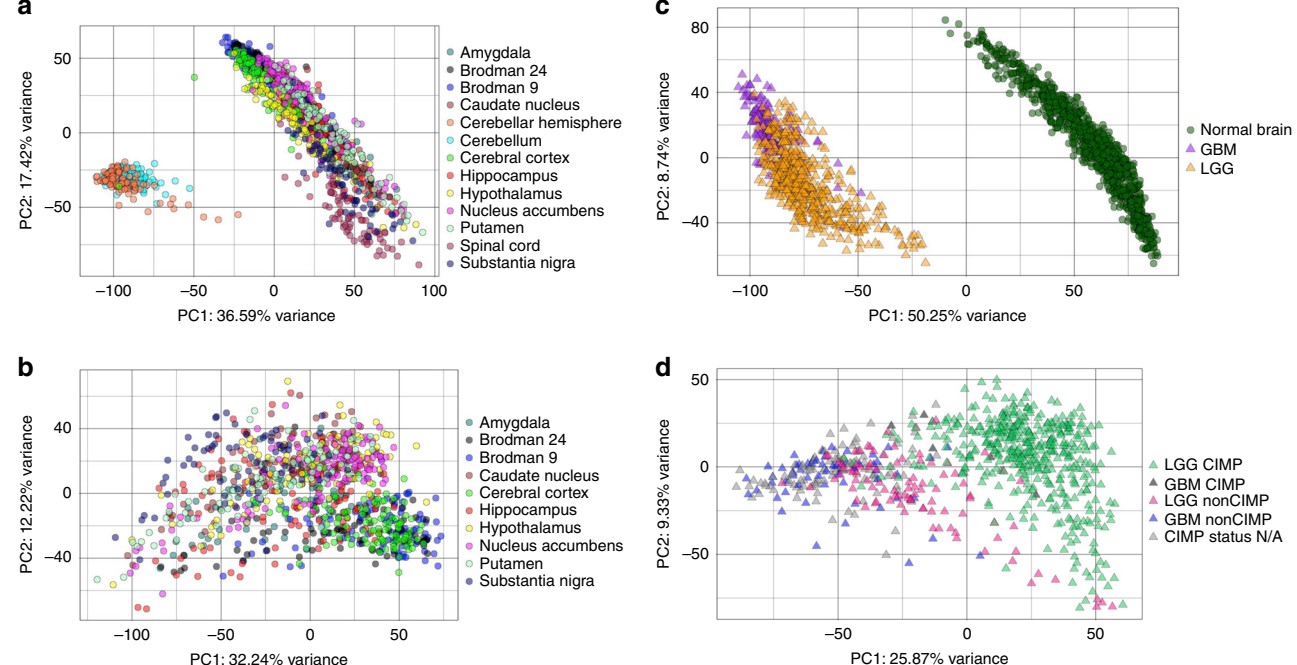

**Fig. 1 Gene expression differences in human normal brain and human glioma. a** Principal component analysis (PCA) plot of all brain samples from Genotype-Tissue Expression (GTEx) project. Samples from 13 regions fall into two groups clearly separating the cerebellum samples from the rest of the brain regions. **b** Principal Component analysis (PCA) plot of pooled brain samples from GTEx (with cerebellum and spinal cord removed) and The Cancer Genome Atlas (TCGA) samples using gene expression from 20,214 genes from hg19. Two clear clusters were formed, TCGA-GBM and TCGA-LGG samples fall into one cluster while the supratentorial regions from GTEx maintain the one pooled normal brain cluster from **a**. **c** PCA plot of pooled normal brain regions broken down shows minimal regional specificity and clustering based on 20,214 genes from hg19. **d** PCA plot showing tumor samples from TCGA using gene expression of 20,214 genes in hg19 colored by CpG Island Methylation Phenotype (CIMP) status. CIMP and non-CIMP status for TCGA samples were obtained from Bolouri et al.[26], and show distinct CIMP vs non-CIMP clusters based on total gene expression. Normal brain regions are displayed as circles, brain tumors are displayed as triangles.

**Table 2 log2(TPM) values for *NTRK2* whole gene expression, TrkB.FL transcript expression, and TrkB.T1 transcript expression.**

|  | GBM | LGG | Normal brain |
| --- | --- | --- | --- |
| Whole gene *NTRK2* |  |  |  |
| Minimum | 0 | 0.7075 | 1.268 |
| First quartile | 2.2741 | 5.957 | 5.378 |
| Median | 3.6168 | 7.0254 | 6.122 |
| Mean | 3.6745 | 6.7596 | 5.964 |
| Third quartile | 5.2447 | 7.9051 | 6.655 |
| Maximum | 8.1451 | 10.5174 | 8.447 |
| TrkB.FL transcript |  |  |  |
| Minimum | 0 | 0 | 0 |
| First quartile | 0.8538 | 2.151 | 0.8105 |
| Median | 1.2465 | 2.919 | 1.3802 |
| Mean | 1.4758 | 2.906 | 1.4481 |
| Third quartile | 2.0373 | 3.741 | 1.9504 |
| Maximum | 4.5814 | 5.701 | 4.2866 |
| TrkB.T1 transcript |  |  |  |
| Minimum | 0.6939 | 0 | 0 |
| First quartile | 3.5196 | 5.679 | 0.08282 |
| Median | 4.6364 | 6.727 | 1.0323 |
| Mean | 4.7139 | 6.426 | 1.48244 |
| Third quartile | 5.8816 | 7.611 | 2.69518 |
| Maximum | 8.0924 | 10.084 | 6.11901 |

**TrkB.T1 expression is increased in human gliomas.** Given TrkB's diverse roles in neurodevelopment, we first sought to compare the expression of the two most studied *NTRK2* splice variants[17,28,29], the full-length receptor tyrosine kinase (TrkB.FL),

and kinase-deficient truncated isoform (TrkB.T1). Using publicly available transcript data from GTEx (available as RPKM data) and TCGA (available as RSEM counts from the legacy archive), we converted all transcript data to Transcripts Per Million (TPM) to allow for isoform comparisons across TCGA and GTEx[30]. Contrary to existing hypotheses surrounding the full-length kinase, TrkB.FL, as the sole suspected *NTRK2* contribution to oncogenesis, TrkB.FL levels remain relatively consistent across pooled normal supra-tentorial regions, LGG and GBM (Fig. 2a, Table 2, Supplementary Data 2). Moreover, in contrast to a suspected role of the TrkB kinase in gliomas, we found that high transcript expression of TrkB. FL is associated with *better* prognosis for both GBM and LGG, not worse prognosis (Supplementary Fig. 1). By contrast, transcript expression levels of kinase-deficient TrkB.T1 were significantly increased in both CIMP and non-CIMP gliomas (LGG and GBM) compared with all normal brain regions (Fig. 2a, Table 2). Compared with TrkB.FL, TrkB.T1 emerged as the predominant isoform expressed in nearly all human gliomas in TCGA (Fig. 2b). Further analysis across all *NTRK2* isoforms confirmed that transcripts with the unique TrkB.T1-specific 11-amino acid C-terminus pre-dominate not only over transcripts containing the TrkB kinase, but also when compared with all other NTRK transcripts (including *NTRK1*, *NTRK2*, *NTRK3*) (Supplementary Data 2). Analysis of 50 human glioblastoma stem cell (GSC) lines isolated from primary tumors show that this TrkB.T1 variant predominates over all other NTRK isoforms[31], further highlighting its potential role in brain tumor biology (Fig. 2c and Supplementary Fig. 2, Supplementary Data 3).

**TrkB.T1 distribution is altered in neoplastic brain.** As TrkB.T1 is the predominantly expressed isoform compared with TrkB.FL

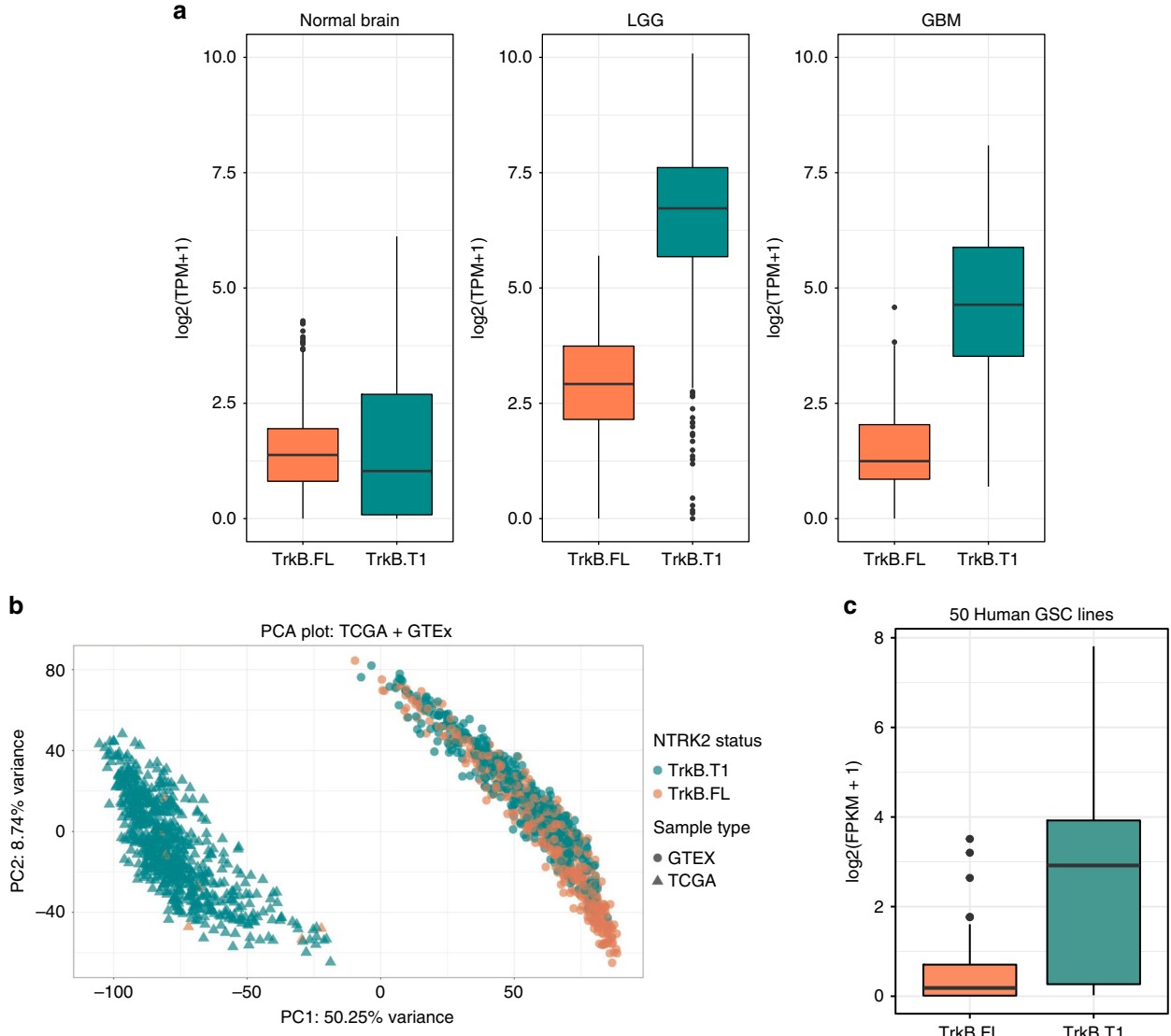

**Fig. 2 *NTRK2* splice variant, TrkB.T1 is the predominant TrkB isoform, in human glioma. a** Boxplots for transcript expression of TrkB.FL and TrkB.T1 from brain samples from GTEx project (n = 956) and TCGA (n = 530) LGG and TCGA GBM (n = 166) demonstrate a predominance of TrkB.T1 in glioma samples compared with normal brain. TrkB.T1 transcript displayed in teal, TrkB.FL transcript displayed in orange. **b** Principal Component analysis (PCA) plot of all brain samples from GTEx project and TCGA samples using gene expression from 20,214 genes from hg19. Samples are colored based on status of predominant *NTRK2* transcript: teal for TrkB.T1 and orange for TrkB.FL. Normal brain regions are displayed as circles, brain tumors are displayed as triangles. **c** TrkB.T1 and TrkB.FL transcript expression shows increased expression of TrkB.T1 compared with TrkB.FL across 50 human glioblastoma stem cell (GSC) lines (6 lines: BTSC349, BTSC349, h543, h516, h561, h676; 44 lines from ref. [31]) (n = 50 for each isoform (TrkB.T1 and TrkB.FL); t-test, p = 1.9 × $10^{-08}$). For **a** and **c**, data are represented as boxplots where the middle line is the median, the lower and upper hinges correspond to the first and third quartiles (the 25th and 75th percentiles), the upper whisker extends from the hinge to the largest value no further than 1.5 * IQR from the hinge (where IQR is the inter-quartile range, or distance between the first and third quartiles) and the lower whisker extends from the hinge to the smallest value at most 1.5 * IQR of the hinge while data beyond the end of the whiskers are outlying points that are plotted individually[85].

in LGG and GBM, we used differential gene correlation analysis (DGCA)[32] to find differentially correlated genes whose expression was positively correlated with *NTRK2* in LGG or GBM and anti-correlated with *NTRK2* in normal brain. Of these significantly correlated genes, the top 350 genes for each tumor type were subjected to gene ontology (GO) analysis to determine which, if any, classes of genes were enriched in *NTRK2* glioma pairings (Fig. 3a). GO analysis for both biological processes and cellular component revealed a predominance of genes implicated in morphogenesis and proliferation, as well those implicated in endocytic and vesicular transport (Fig. 3a–c and Supplementary Data 4), suggesting that the subcellular location of TrkB.T1 may be critical to its function and could be different between tumor

and normal brain. Additionally, because previous work has alluded to more efficient recycling of TrkB.T1 receptors back to the plasma membrane in PC12 cells and neurons[33], we next wanted to identify the distribution of TrkB.T1 in normal human brain and human gliomas based on these DGCA results implicating endocytic and vesicular transport genes.

Basic scientific and clinical investigation surrounding TrkB's role in neurodevelopment and oncology has often been hindered due to its complex splicing patterns combined with frequent inability of available antibodies to distinguish between TrkB isoforms. While early immunohistochemical analyses of neural tumors using pan-Trk antibodies confirmed presence of at least one neurotrophin receptor[34], little insight could be gained as to

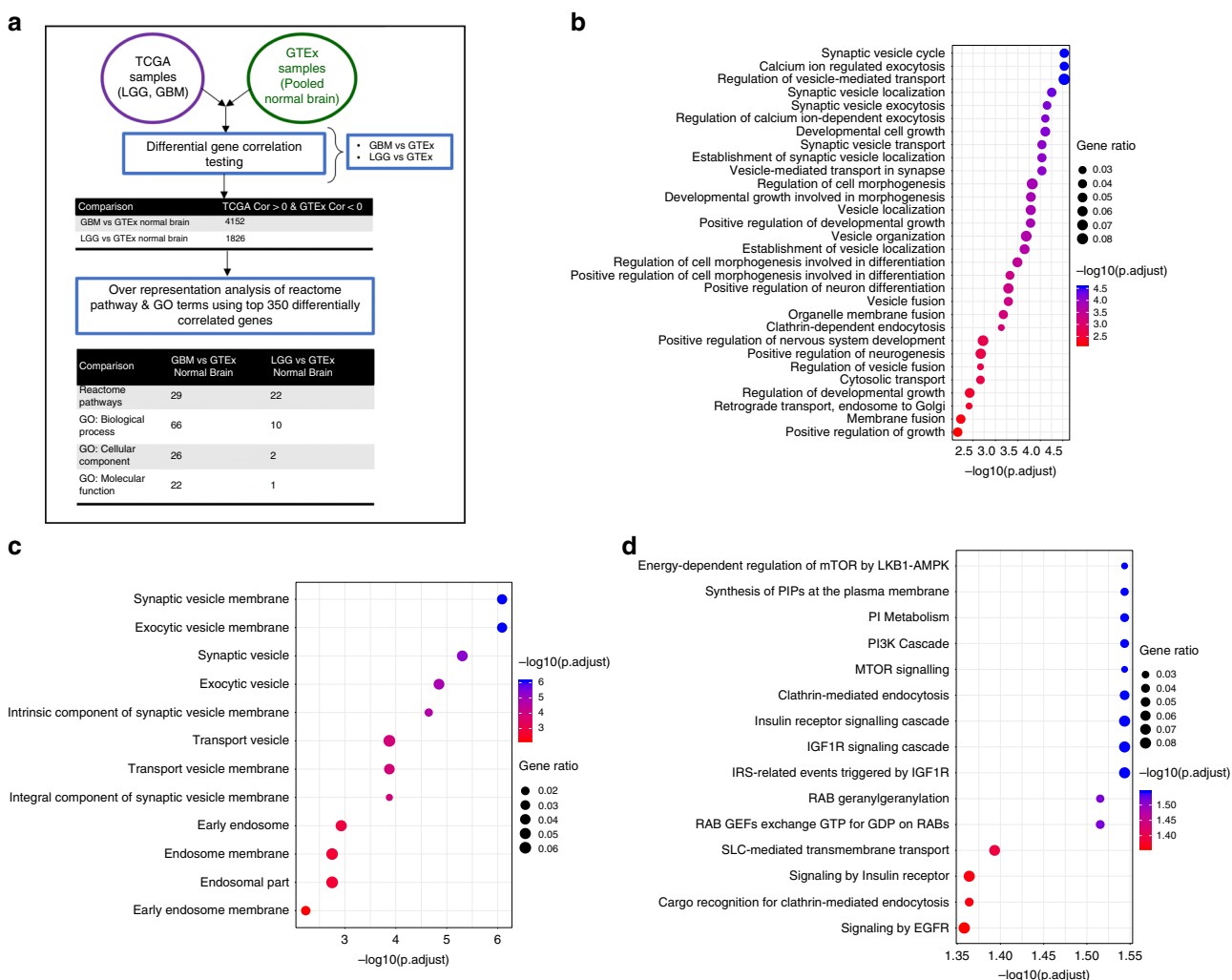

**Fig. 3 GO enrichment analysis for *NTRK2* in GBM or LGG. a** Schematic of DGCA analysis for the top 350 genes significantly positively correlated with *NTRK2* in LGG or GBM. GO terms for biological processes in top 350 differentially expressed genes in GBM compared with normal brain (**b**) and GO terms for the top 350 differentially expressed genes in the cellular component in GBM compared with normal brain (**c**) revealed a predominance of genes implicated in morphogenesis and proliferation, as well endocytic compartments and vesicular transport. **d** Gene ontology enrichment terms for the genesets within the reactome pathways in top 350 differentially expressed genes in LGG compared with normal brain.

which TRK (or TRKs) were present or if receptor distribution differed between neural and non-neural tissue. Further, TrkB-specific antibodies do not easily discriminate between the full and truncated variant gene products as the majority are generated against either the entire extracellular domain or extracellular subdomains—regions conserved between full-length and various truncated isoforms[35] (Supplementary Table 1). This lack of reagent specificity does not prohibit certain assays but has made visualization of endogenous TrkB splice variants difficult and often requires the use tagged constructs or in vitro systems. To determine endogenous localization of TrkB.T1 in human brain, we designed, developed, and validated a monoclonal antibody to the unique intracellular region of TrkB.T1's 11-conserved amino acids (FVLFHKIPLDG) through a series of peptide fusions and immunogenic boosting of TrkB.T1$^{-/-}$ mice as strong cross-species conservation of this receptor has previously made standard rabbit or mouse colonies unsuitable for successful antibody production[36] (Fig. 4a and Supplementary Fig. 3).

Normal brain tissue from rapid autopsies ($n = 5$) was procured and subjected to TrkB.T1 immunohistochemistry (IHC) with this TrkB.T1-specific antibody. Each control case contained both supratentorial and infratentorial brain regions, including cerebral cortex, cerebellum, hippocampus, subiculum, entorhinal cortex, and thalamus (Fig. 4b and Supplementary Fig. 4a, b). TrkB.T1 IHC on normal human brain reveals a distinct punctate intracellular localization and distribution of TrkB.T1 throughout the frontal cortex, hippocampus, and cerebellum. Several normal brain regions exhibit abundant neurons, and glial cells, with strongly intense, cytoplasmic TrkB.T1 immunohistochemical staining in a predominantly vesicular pattern (Supplementary Fig. 4a, b).

Predominant TrkB.T1 expression in gliomas (Fig. 2, Supplementary Data 2) combined with GO enrichment analysis data showing altered genesets involved in endocytic trafficking in brain tumors compared with normal brain (Fig. 3) suggested that TrkB.T1 localization may be altered in LGG and GBM. We next sought to compare the distribution and localization of TrkB.T1 in brain tumors compared with normal brain tissue. Formalin-fixed paraffin-embedded (FFPE) pathology tissue was reviewed by a neuropathologist and characterized according to three major integrated classifications of diffuse gliomas derived from the 2016 World Health Organization (WHO) classification of CNS tumors: Oligodendroglioma, IDH-mutant and 1p/19q-codeleted ($n = 5$); Astrocytoma/Glioblastoma, IDH-mutant ($n = 5$); and

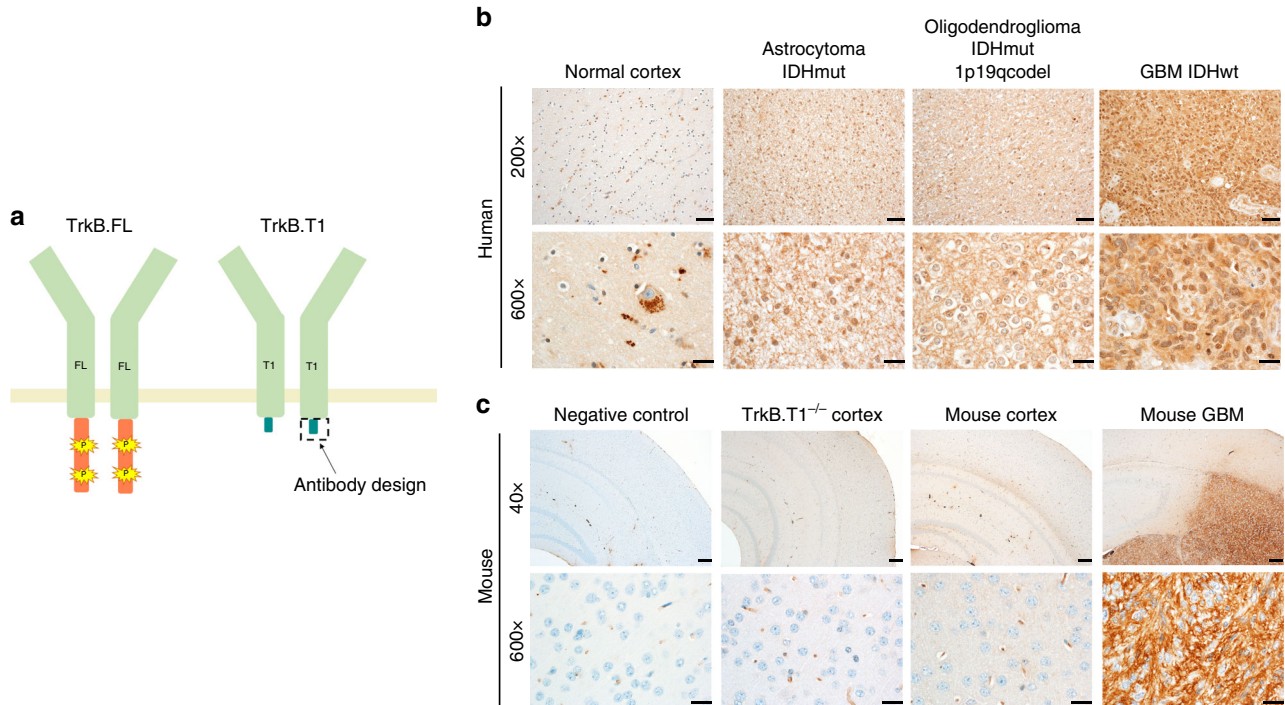

**Fig. 4 Human and rodent TrkB.T1 immunostaining in normal brain and glioma. a** Receptor schematic of TrkB.FL (kinase domain demonstrated in orange with phosphorylation sites in yellow) and TrkB.T1 highlighting region of antibody specificity (11-amino acid region shown in teal blue). Extracellular domain, transmembrane domain, and first intracellular amino acids common to both variants shown in light green. TrkB.T1 immunohistochemistry shows punctate vesicular staining in normal human (**b**) and mouse cortex (**c**) compared with intense diffuse staining in human glioma (**b**) and mouse glioma (**c**) with lack of punctate vesicular pattern. Lack of staining in rodent negative control (no primary antibody) and TrkB.T1$^{-/-}$ cortex demonstrates antibody specificity for TrkB.T1 splice variant. (IDHmut—IDH mutant; 1p19q codel—1p 19 q co-deleted). Immunohistochemistry was performed on independent biological samples of each tumor type, in replicates of 3–5. Representative images were chosen and additional images are shown in Supplementary Fig. 4–6. Photomicrographs are as specified at 40×(scale bar = 200 μm), 200×(scale bar = 50 μm), and 600×(scale bar = 20 μm).

Astrocytoma/Glioblastoma, IDH-wild-type (WT) ($n = 14$ total; $n = 5$ classical, $n = 4$ proneural, $n = 5$ mesenchymal)[37,38]. Pathological review of tumor sections confirms that TrkB.T1 staining is intensely diffuse throughout all CIMP and non-CIMP tumors. Specifically, these tumors lack punctate intracellular clustering of the receptor seen in non-neoplastic cells, demonstrating that both the expression and distribution of TrkB.T1 differ in normal brain compared with gliomas, providing a degree of visual specificity and receptor localization that was previously not possible. Diffuse gliomas of all classifications demonstrated patchy to widespread, moderately intense, cytoplasmic TrkB.T1 immunohistochemical staining with noticeable lack of normative vesicular pattern (Fig. 4b and Supplementary Fig. 4c).

TrkB.T1 IHC on normal mouse cortex recapitulated the cytoplasmic vesicular staining found in normal human cortex and this pattern was completely absent in TrkB.T1$^{-/-}$ brain (Fig. 4c, Supplementary Fig. 5a). We then stained archived samples of several different RCAS/tv-a mouse models of glioma including those driven by PDGF and those driven by loss of combined NF1 and PTEN or EGFRvIII[39]. All mouse glioma models analyzed, regardless of strain, genetic background, or oncogenic driver showed strong diffuse staining for expression of endogenous TrkB.T1 within the tumor boundaries (Fig. 4c and Supplementary Fig. 5a, b), suggesting that TrkB.T1 may be selected for as the predominant *NTRK2* isoform across multiple tumor types. These patterns observed in normal mouse brain and mouse gliomas match those seen in normal human brain and human glioma described above (Fig. 4c and Supplementary Fig. 4). As an internal control, TrkB.T1 immunostaining was negative in human glioma stromal blood vessels (Supplementary Fig. 4c).

and TrkB.T1$^{-/-}$ mouse cortex (Fig. 4c and Supplementary Fig. 5a). These tumors also showed little to no staining for TrkB.FL when stained with an antibody developed against amino acid 810 of the TrkB kinase domain, which is in contrast to normal mouse cortex and TrkB.T1$^{-/-}$ cortex where TrkB.FL is present at moderately high levels (Supplementary Fig. 6). To ensure IHC on mouse tumors was not an artifact of mouse-on-mouse reagents, we also created a recombinant antibody with a rabbit FC region that could be detected with anti-rabbit secondary antibodies, which showed similar IHC results (Supplementary Fig. 4b).

**TrkB.T1 enhances PDGF-driven gliomagenesis in vivo.** Given the differences in TrkB.T1 expression and distribution in normal brain and gliomas in both humans and mice, we sought to explore the role of TrkB.T1 in glioma biology utilizing the RCAS/ tv-a system, which allows somatic transfer TrkB.T1 into specific cell types of genetically altered mice[40,41]. RCAS-TrkB.T1 injected into the brains of postnatal day (P)1 *N/tv-a* (Nestin-expressing cell of origin) wild-type mice significantly enhanced glioma formation as evidenced by decreased survival in mice injected with RCAS-PDGFB + RCAS-TrkB.T1 compared with mice injected with RCAS-PDGFB only (RCAS-PDGFB + RCAS-TrkB.T1 vs. RCAS-PDGFB alone: median survival 34 days vs. 109 days, Mantel-Cox/log-rank hazard ratio 2.306, 95% confidence interval 0.8999 to 5.909, $p < 0.05$) (Fig. 5a). In addition to strong immunostaining of TrkB.T1 across genetically diverse RCAS-driven gliomas (Supplementary Fig. 5b), additional analysis of genetically diverse mouse tumorsphere lines reveals increased expression of TrkB.T1 compared with TrkB.FL (Supplementary Data 3,

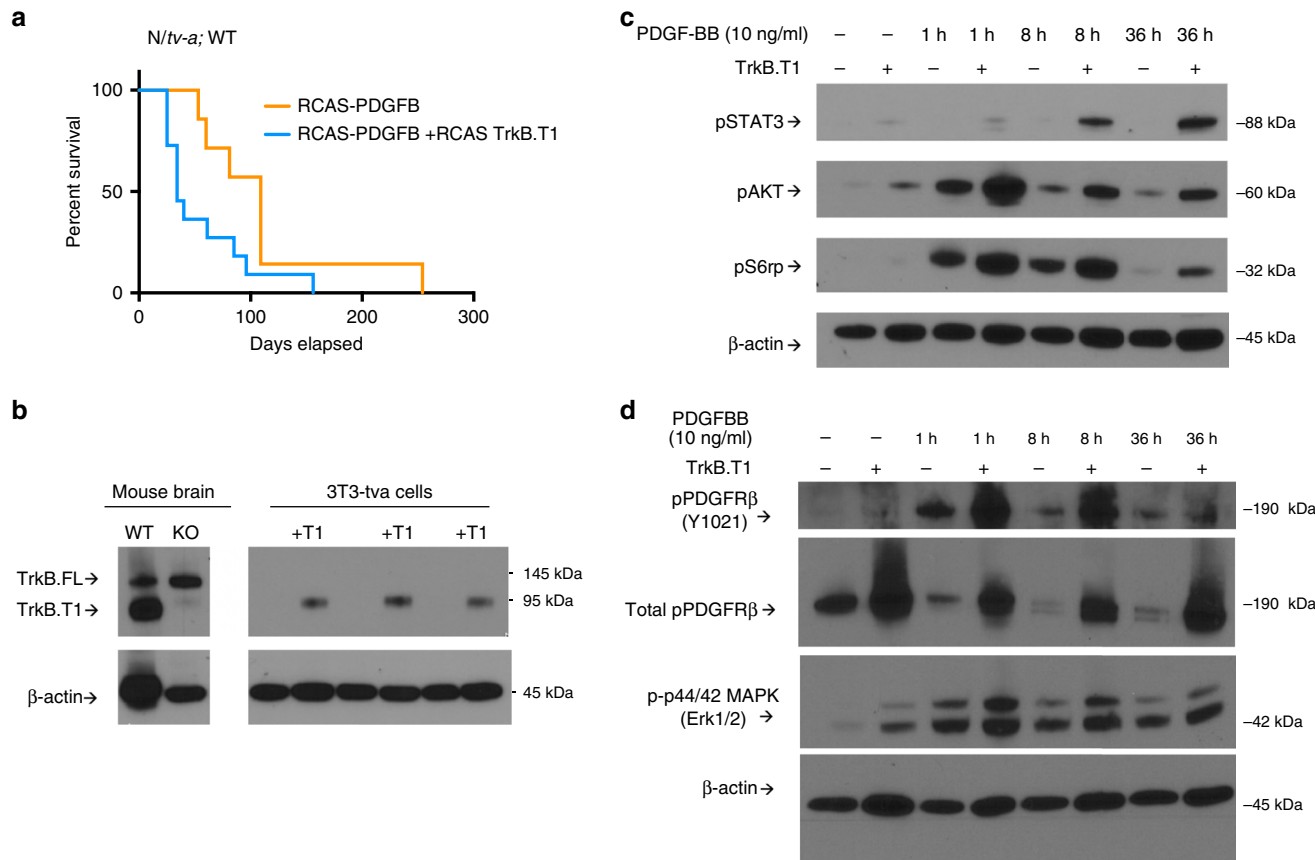

**Fig. 5 TrkB.T1 enhances PDGF signaling in vivo and in vitro. a** Kaplan–Meier plot showing symptom-free survival of PDGFB (orange) vs PDGFB + TrkB.T1 (blue) induced gliomas (median survival 34 days vs 109 days; Mantel-Cox/log-rank hazard ratio 2.306, 95% confidence interval 0.8999 to 5.909, $p =$ 0.0452. **b** Western blot showing representative efficiency of RCAS-TrkB.T1 in 3T3/tv-a cells (lanes 4, 6, 8) in relation to normal mouse brain (lane 1) and TrkB.T1$^{-/-}$ brain control (lane 2). **c** Representative western blot showing TrkB.T1-expressing 3T3/tv-a cells exhibit enhanced pSTAT3, pAKT, pS6rp, pERK signaling in response to 10 ng/ml PDGF-BB treatment over the course of 36 h; SS (serum starved), with beta-actin loading control; 3T3/tv-a PDGF-BB treatment time-points repeated in duplicate or triplicate. **d** TrkB.T1-expressing 3T3/tv-a cells show enhanced phosphorylation of PDGFR-β (Y1021) and maintain expression of total PDGFR-β upon ligand stimulation. Western blotting was performed in triplicate on biologically independent samples, in duplicate using technical replicates, and representative images were chosen.

Supplementary Fig. 7a, b) suggesting that while overexpression TrkB.T1 can enhance PDGF-driven gliomas in vivo, this variant also predominates over TrkB.FL in all rodent gliomas analyzed and may be selected for in an oncogenic context.

Our *NTRK2* transcript-level analysis of normal brain and human gliomas reveals relatively low amounts of full-length TrkB.FL compared with abundant amounts of TrkB.T1 in tumor tissue, suggesting that this isoform has an important role in both normal neurobiology and oncogenesis alike. Subcellular distribution patterns of this TrkB.T1 suggest potentially distinct roles for neuronal and glial cell populations in normal brain, whereas as the lack of vesicular pattern in CIMP and non-CIMP tumors highlights the need for further characterization of differences in signaling mechanism as a result of receptor compartmentalization. As mentioned above, in addition to endocytic and vesicular compartment GO terms (as shown in Fig. 3 and Supplementary Data 4), DGCA GO analysis for positive *NTRK2* correlations in glioma reactome pathways revealed enrichment for genes in the PI3K and mTOR signaling pathways (Fig. 3d and Supplementary Data 4), which have been known to regulate cellular proliferation in normal and oncogenic contexts. Previous work has shown that TrkB.T1 enhances both apoptosis and proliferation of neurospheres leading to larger sphere diameter[42] and RCAS-TrkB.T1 delivery to *N/tv-a* isolated neurospheres recapitulates this phenomenon as shown by the increased size of neurospheres

infected with RCAS-TrkB.T1 compared with control neurospheres (Supplementary Fig. 7c) suggesting that the TrkB.T1 isoform can influence the replicative capacity of progenitors in dual systems.

**TrkB.T1 enhances perdurance of PDGF-induced signaling**. To further characterize which signaling pathways may be active in TrkB.T1 gliomas, 3T3/tv-a cells were maintained as previously described[39] and infected with RCAS-TrkB.T1 (Fig. 5b) to explore potential mechanistic links for these effects. These cells were chosen for their lack of endogenous TrkB receptor expression so that effects of TrkB.T1 could not be confounded with potential interaction with TrkB.FL (see Fig. 5b and ref. [43]). In order to explore if TrkB.T1 can enhance growth factor signaling in vitro, 3T3/tv-a cells were treated with PDGF-BB ligand, chosen for its role in glioma biology, for varying durations and lysates were subjected to western blot analysis to explore downstream targets often implicated in gliomagenesis. TrkB.T1-expressing 3T3/tv-a cells showed significantly enhanced and sustained levels of phospho-STAT3, phospho-AKT, phospho-S6rp, and slight increases in phospho-ERK in response to PDGF treatment (Figs. 5c, d), suggesting a role for TrkB.T1 in enhancing perdurance of certain components of PDGF signaling, potentially through altered endocytic trafficking, tyrosine kinase receptor

crosstalk, sustained membrane insertion or recycling of PDGFR-β. TrkB variants, in general, have been shown to alter default recycling pathways[33] and exhibit cross-talk with other tyrosine kinases in different cancer types[44], suggesting that interactions with other signaling cascades have yet to be elucidated.

While rapid downregulation of PDGFβ receptors has been observed upon ligand activation[45], similar downregulation of transactivated PDGFβ receptors has not[46]. Furthermore, transactivation of PDGF-β and TrkB receptors has been shown to occur in unique contexts, such as in neuroblast cells under low concentrations of reactive oxygen species (ROS)[47]. Western blot analysis of phosphorylated (p)PDGFR-β and total PDGFR-β demonstrate that there is enhanced pPDGFR-β in TrkB.T1-expressing 3T3/tv-a cells and also a maintained expression of total PDGFR-β upon ligand stimulation (Fig. 5d). Future biochemical studies characterizing the precise role of TrkB.T1 in the recycling, stabilization, or degradation PDGFR-β will shed light on how this interaction achieves the downstream signaling effects shown in Fig. 5c and in vivo effects observed in Fig. 5a.

**GO analysis of GSCs stratified by TrkB.T1 expression.** To explore the correlations between TrkB.T1 levels and GO terms in human tumor initiating GBM stem cells (GSCs), we took whole gene expression data from four previously published[48,49] human GSC lines derived from patient tumors. Two classical/proneural lines were chosen—GSC line 559 was defined as TrkB.T1 Low with TrkB.T1 transcript FPKM values of 1.86, 0.84, and 0.81 for triplicate wells, while GSC line 448 was defined as TrkB.T1 High with FPKM values of 74.94, 58.80, 45.68. We found differentially expressed genes between these two classical/proneural cell lines (Fig. 6a, b) (545 significantly upregulated genes and 704 significantly downregulated genes, adjusted p-value (FDR) < 0.05 with a fold change of more than 2 in each respective direction) (See Supplementary Data 2 and Supplementary Data 5). To explore potential pathways involved with high and low TrkB.T1 expression in vitro, GO analysis was performed which revealed downregulated cellular compartment genesets involved in coated vesicle membranes, clathrin-coated vesicle membranes, clathrin coated endocytic vesicle membranes, endocytic vesicle membrane, and clathrin-coated endocytic vesicles in the TrkB.T1 High line (488) compared with TrkB.T1 Low line (559) (Fig. 6c, d, Supplementary Data 5) suggesting endocytic vesicles and receptor trafficking may be altered in TrkB.T1 High vs TrkB.T1 Low contexts. Because normal human brain exhibits relatively lower TrkB.T1 levels, and more punctate TrkB.T1 staining, compared with both GBM and LGG (Fig. 2), these differentially expressed GSC genesets are compatible with the differences in TrkB.T1 distribution shown in Fig. 4. GO analysis also revealed downregulated molecular function genesets in TrkB.T1 High (488) compared with TrkB.T1 Low (559) GSC lines involved in platelet-derived growth factor binding and transcriptional activity, further highlighting the potential role for TrkB.T1 PDGF-regulated signaling (as shown in Fig. 5). To confirm that TrkB.T1 specific terms were not specific to classical/proneural lines, we also separately analyzed a second pair of classical/mesenchymal GSC lines. (Fig. 6a, b)—GSC line G179 was defined as TrkB.T1 Low with TrkB.T1 transcript FPKM values of 2.12, 1.70, and 1.50 for triplicate wells, while GSC line G14 was defined as TrkB.T1 High with FPKM values of 41.78, 37.48, 34.38. We found differentially expressed genes between these two classical/mesenchymal cell lines (Fig. 6a, b) (678 significantly upregulated genes and 1032 significantly downregulated genes, adjusted p-value (FDR) < 0.05 with a fold change of more than 2 in each respective direction) (See Supplementary Data 5).

**TrkB.T1 upregulates PI3K/Akt pathway genes in neural stem cells (NSCs).** To explore the causal role of TrkB.T1 in normal cells, we cloned TrkB.T1, TrkB.FL, and GFP into pLJM1 lentiviral vectors to infect NSCs[50,51] and performed RNA-sequencing. TrkB.FL-transduced NSCs showed increased levels of TrkB.FL RNA, and protein as validated by western blot (Supplementary Fig. 9a). TrkB.T1 transduced NSCs showed increases in TrkB.T1 RNA and protein and also exhibit increases in gene signatures previously characterized as a marker of a cell's proliferative index[52,53], compared with TrkB.FL (Supplementary Fig. 9b). We found that while each infected line clustered independently (Fig. 7a, b), differential expression analysis revealed that 17 genes were significantly upregulated in TrkB.T1 infected NSCs compared with TrkB.FL infected NSCs (adjusted p-value < 0.05 and fold change more than 25%) (Fig. 7c and Supplementary Data 6). GO enrichment analysis on these upregulated genes revealed a predominance of terms in the PI3K/Akt pathway and PI3K/ERBB2/ERBB4 (Fig. 7c, d and Supplementary Fig. 8), suggesting that while TrkB.T1 is indeed capable of enhancing PDGF-induced Akt signaling as shown above (Fig. 5), it may also have more general effects on basal PI3K/AKT signaling in the absence of exogenous ligand, highlighting the generalizability of TrkB.T1 – PI3K/AKT interactions across various cell types.

Treatment of NSCs transduced with TrkB.FL and TrkB.T1 showed that both lines were sensitive to phosphoinositide-3 kinase (PI3K) and mTOR inhibitors (LY294002 and rapamycin, respectively) in a dose dependent manner and that TrkB.T1 NSCs were marginally, but statistically significantly, more sensitive to LY294002 than TrkB.FL NSCs at both 48 and 72 h (Supplementary Fig. 8d, 8e). Combined with RNASeq data suggesting a role for TrkB.T1 in PI3K signaling and enhancement of downstream PDGFR pathways in vitro (Figs. 5, 7, Supplementary Fig. 8, Supplementary Data 6), increased sensitivity to LY294002 in the TrkB.T1 NSCs suggests that TrkB.T1 may selectively activate PI3K pathways.

Similar to TCGA DGCA and GSC data, there were also significant differences in gene expression between TrkB.T1 and TrkB.FL NSCs for genes involved in endocytic vesicles, vesicular trafficking and vesicular transport (Supplementary Fig. 8 and Supplementary Data 6). TrkB.T1 NSCs also exhibited downregulation of several MHC Class II genes, including HLA-DRB1, HLA-DRB5, HLA-DMA, HLA-DRA1, as well as the master transcriptional activator controlling expression of MHC Class II genes CITTA (Figs. 7c, d, Supplementary Data 6), highlighting the possibility that along with enhancing PDGF-driven signaling in vivo, the increased levels of TrkB.T1 observed in gliomas (Fig. 2) may also harbor a role in modulating antigen-specific immune responses that are often dysregulated cancer.

## Discussion

*NTRK2*'s suspected oncogenic role has long centered around the sole possibility of constitutive activation of the TRK kinase, which has more recently been highlighted in *NTRK2* kinase-activating fusions. While current TRK inhibitors are being developed and being used clinically with some success, it is unclear of the overall specificity of these compounds and if the targeted kinases are being activated independently or through more paracrine mechanisms. The data shown here reveal a kinase-deficient isoform, TrkB.T1, to be the predominant isoform in brain tumors compared with normal brain, which is contrary to the notion that the TrkB's own kinase is the sole oncogenic driver. While the specificity of the 11 exons comprising TrkB.T1's C-terminus is unclear, it remains conserved across species at the amino acid level, highlighting the possibility for an evolutionarily selected function on its own or through interaction with nearby receptor

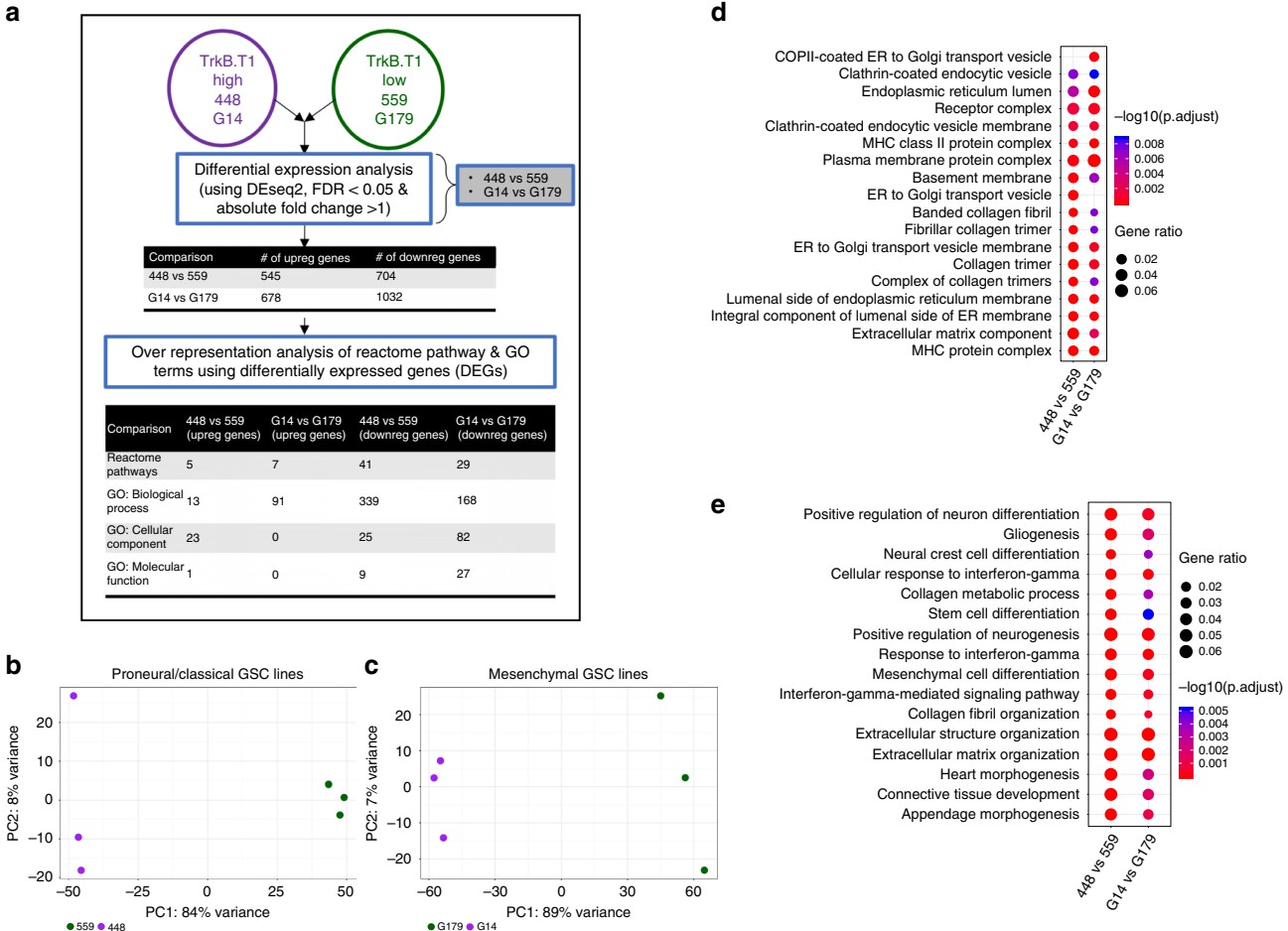

**Fig. 6 DE analysis reveals distinct gene expression patterns between TrkB.T1 High vs TrkB.T1 Low GSC lines. a** Schematic for differential gene expression analysis between TrkB.T1 High (classical/proneural replicates 448 1–3; classical/mesenchymal G14 replicates 1–3) and TrkB.T1 Low (classical/proneural replicates 559 replicates 1–3; classical/mesenchymal G179 replicates 1–3). **b** PCA plots show distinct clusters for the TrkB.T1 High GSC line (488) compared with the TrkB.T1 Low GSC (559) line in the classical/proneural lines and distinct and (**c**) for the TrkB.T1 High GSC Line (G14) compared with the TrkB.T1 Low GSC line (559). GO analysis revealed distinct genesets downregulated in the cellular compartment (**d**) and distinct genesets downregulated in biological processes (**e**) in the TrkB.T1 High (448 and G14) line compared with the TrkB.T1 Low (559 and G179) lines. TrkB.T1 Low lines (559 and G179) shown in green and TrkB.T1 High lines (448 and G14) shown in purple.

tyrosine kinases. Our in-depth transcript level analyses suggest additional components of *NTRK2* biology may be implicated in gliomas. TrkB.T1 exhibits distinct granular and vesicular patterning in normal human and mouse brain and a more diffuse, mis-localized pattern in neoplastic brain (in both mice and humans). In addition to altered receptor localization, TrkB.T1 is not only elevated in human gliomas, but also in murine gliomas driven by multiple genetic mechanisms, and across human and rodent GSC and tumorsphere lines (Supplementary Figs. 2 and 7). Our data show that while PDGF drives tumors with elevated levels of endogenous TrkB.T1, forced expression of TrkB.T1 makes these tumors more aggressive. TrkB.T1 is also capable of enhancing downstream targets of PDGF signaling in vitro, including Akt, STAT3, and pS6 (Fig. 5), and future studies should explore if this is a PDGF-specific effect or a more general effect of TrkB.T1 exhibiting crosstalk with other receptor tyrosine kinases to explore whether this phenomenon may be applicable to other tumor types beyond gliomas. Next generation sequencing of human NSCs reveals that TrkB.T1 overexpression upregulates genes in PI3K/Akt pathways and PI3K events in ERBB2 (HER2)/ERBB3/ERBB4 signaling (Figs. 7c, d) suggesting that TrkB.T1 may have more general effects on PI3K/Akt biology than just amplifying Akt activity resulting from stimulation with PDGF

ligand. Future work should investigate the role of TrkB.T1 in PI3K/Akt pathways and PI3K/ERBB2/ERBB3/ERBB4 networks across various cancer types, as western blots (corresponding to RNASeq data) show upregulation of ERBB3 receptor in TrkB.T1 transduced NCS, along with increased expression NRG2, a ligand for ERBB3 and ERBB4[54] (Supplementary Fig. 9c).

Despite recent advances in diagnostics and characterization of glioma subtypes, survival rates for certain cancers, such as GBM, remain extremely poor, highlighting the need for novel therapeutic targets. As bioinformatic tools now offer more complex analytical approaches, it is possible that unique receptor splice variants, such as TrkB.T1, may prove to be crucial for understanding glioma biology, and oncogenic signaling in general. Our data suggest that TrkB.T1 has a unique distribution pattern throughout the normal brain and that shifts in expression and distribution of this splice variant may harbor distinct roles in oncogenesis within and outside the CNS either independently from or in conjunction with a wide array of growth factors, kinases, or even simultaneous *NTRK2* fusions. It is of interest to note that in many cases of novel *NTRK2* gene fusions, the TrkB.T1 isoform is still able to be generated in its full form from the reciprocal cross, if the fusion breakpoint is downstream of the unique TrkB.T1 exon. The possibility exists that in addition to

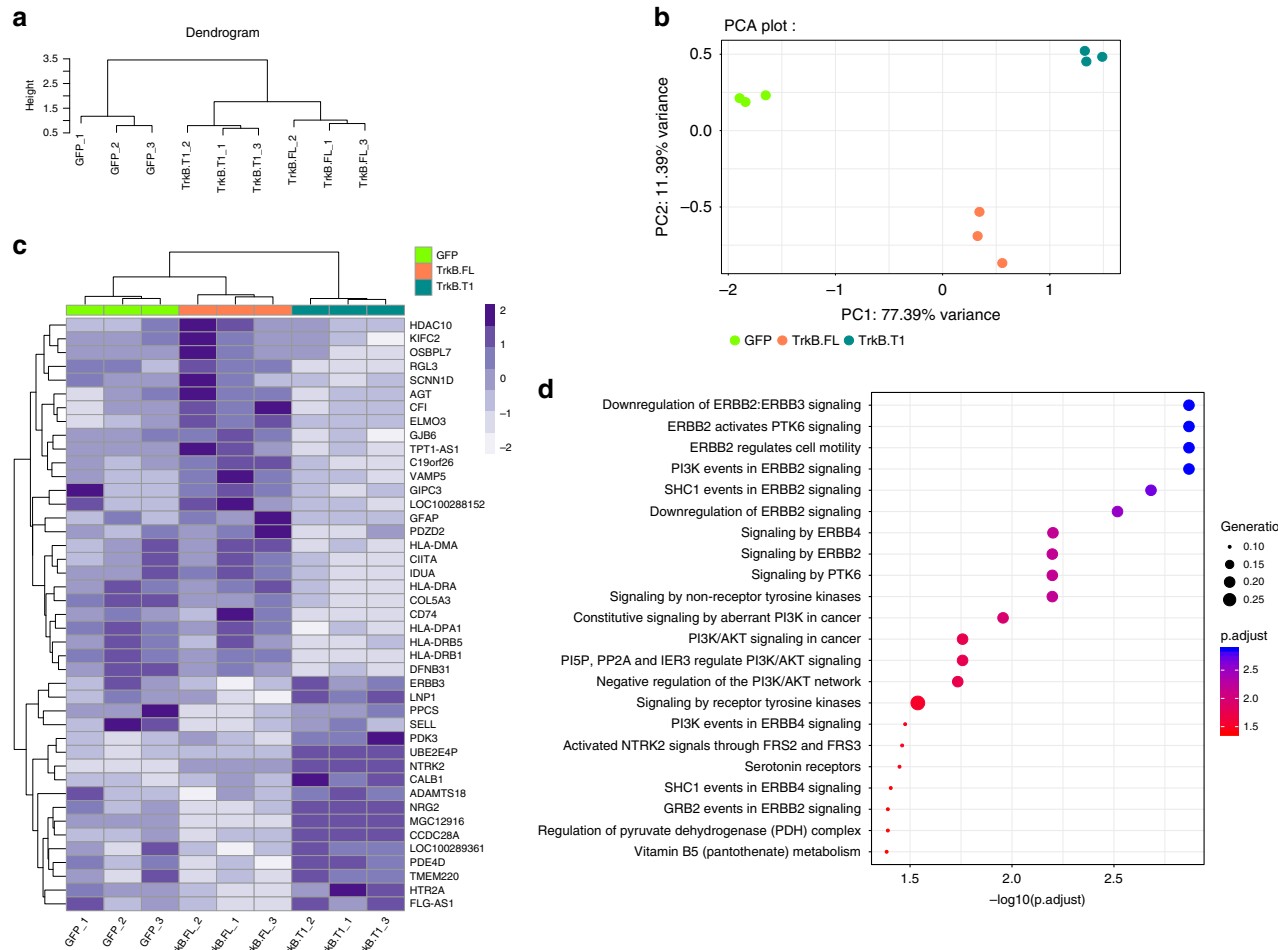

**Fig. 7 TrkB.T1 upregulates genes involved in PI3K/Akt signaling in human NSCs. a** Cluster dendrogram and principal component analysis (PCA) plots (**b**) show that TrkB.T1-transduced, TrkB.FL-transduced, and GFP-transduced NSC lines cluster independently from each other. Heatmap displays differentially expressed genes between TrkB.T1, TrkB.FL, and GFP infected NSCs (**c**). Reactome terms for genes upregulated in TrkB.T1 infected NSCs compared with TrkB.FL infected NSCs revealed genes involved in PI3K/Akt, PI3K/ERBB2/ERBB4 signaling pathways (**d**). GFP-transduced NSCs displayed on PCA plot in bright green, TrkB.T1-transduced cells displayed in teal, TrkB.FL-transduced shows displayed in orange.

creating constitutively active kinases through various fusion partners, these fusions may, in some cases, also be inherently selecting for the production of TrkB.T1[55]. Similarly, analysis of mouse tumorspheres generated with *NTRK2* fusions[56] shows that they express TrkB.T1 as their most highly expressed isoform (Supplementary Fig. 7a, b and Supplementary Data 3). As treatment failure after initial response to TRK inhibitors, such as larotrectinib or entrectinib, is becoming frequently observed with continued use[57], teasing apart the exact roles of specific functional domains of NTRK genes will become crucial for understanding and maintaining clinical efficacy. While the importance of a constitutively active TRK kinase should not be overlooked clinically, it will become increasingly important to tease apart the role of TrkB.T1's role in tumors with- and without-*NTRK2* fusions.

## Materials and methods

**Bioinformatic analysis.** *Obtaining and transforming gene level data.* Publicly available whole gene RPKM read counts for TCGA-LGG and TCGA-GBM were downloaded from UCSC's Xena Browser (https://xenabrowser.net/datapages/? host=https://tcga.xenahubs.net). CIMP and nonCIMP status for TCGA-GBM[58] and TCGA-LGG[59] samples were obtained from Bolouri et al.[26] Whole gene RPKM read counts for The GTEx Project (v6)[60] was obtained from https://gtexportal.org/ (version 6) on 03/30/2017 (dbGaP accession number phs000424.v7.p2). GTEx is supported by the Common Fund of the Office of the Director of the National Institutes of Health, and by NCI, NHGRI, NHLBI, NIDA, NIMH, and NINDS. Whole gene counts from GTEX and TCGA was transformed to log2(rpkm + 1)

using R [(version 3.3.2) (R Core Team (2017). R: A language and environment for statistical computing. R Foundation for Statistical Computing, Vienna, Austria. URL https://www.R-project.org/)].

*Obtaining and transforming transcript level data.* Transcript Data (TCGA RNAseqV2 RSEM data) for TCGA-GBM and TCGA-LGG was downloaded from the legacy archive of TCGA. Transcript data (RPKM data) was also downloaded from GTEx(v6) website. To make the RSEM data from TCGA, and RPKM data comparable, we converted both data sets to TPM (transcripts per million). The RSEM counts from TCGA to TPM counts using the following formula[30]: RSEM can be multiplied by $10^6$. The RPKM transcript data from GTEX was converted to TPM data using the formula[30]: TPM = RPKM/(sum of RPKM over all genes/ transcripts) × $10^6$. For transcript analyses, NTRK2 transcript IDs were manually aligned to confirm sequence homology and are as follows: TrkB.FL (UCSC: uc004aoa.1; Ensembl: ENST00000376213.1, NTRK2_201), TrkB.T1 (UCSC: uc004aob.1; Ensembl: ENST00000395882.1, NTRK2_204).

*RNA Seq analysis for 448 vs 559 and G14 vs G179 GSC cell lines.* Raw gene counts for cell line 448, 559, G14, and G179 were obtained from GEO repository, (GSE89623)[61]. Cufflinks[62] results for TrkB.T1 were used to differentiate the 4 cell lines into TrkB.T1 High (448,G14) and TrkB.T1 Low (559, G179) groups. Differential Gene Expression analysis was performed between TrkB.T1 High and TrkB.T1 Low (448 vs 559, G14 vs G179) cell lines using the R/Bioconductor package DESeq2[63] and a statistical cutoff of FDR < 0.05 and FC > 2 was applied to obtain differentially regulated genes.

*RNA Seq analysis for NSC cell lines.* RNA-seq reads were aligned to the UCSC hg19 assembly using STAR2[64] and counted for gene associations against the UCSC genes database with HTSeq[65]. Differential Gene Expression analysis was performed between GFP vs TrkB.T1 NSCs, GFP vs TrkB.FL NSCs, and TrkB.T1 vs TrkB.FL NSCs, using the R/Bioconductor package edgeR[66] and a statistical cutoff of FDR < 0.05 and FC > 1.25 was applied to obtain differentially regulated genes. Raw data files have been deposited to the NCBI Gene Expression Omnibus under accession number GEO: (GSE136868).

*Downstream analysis.* For PCA plots, the R function 'dist' was used to calculate Euclidean distance between samples. We computed the PCA for the above distances and visualized it using R package ggplot2 (v 2.2.1). Transcript Data was visualized using boxplots using R package ggplot2[67]. Whole gene correlations between Normal Brain Data from GTEX and LGG, GBM data from TCGA was calculated using R package DGCA (https://cran.r-project.org/web/packages/DGCA/index.html). For predominant transcript analysis, log2 TPM values for each transcript were compared and each sample from pooled normal brain (GTeX), LGG (TCGA) and GBM (TCGA) was marked as either TrkB.T1 or TrkB.FL, depending on which transcript, of the two, had the greater value. GO and Reactome Pathway enrichment analysis[68,69] was done using R Bioconductor Packages clusterProfiler v 3.4.4[70] and dot plots were made using R Bioconductor package DOSE[71]. Heatmaps were made using R package pheatmap (https://CRAN.R-project.org/package=pheatmap).

*GSC transcript analysis.* The human GSCs dataset in Fig. 2 and Supplementary Fig. 2 consists of 44 samples from GSE119834[31] and 6 additional lines (BTSC349, BTSC349, h543, h516, h561, and h676) from GSE150653. For the newly generated data, sequencing libraries were prepared with the NEBNext Ultra II Directional RNA Library Prep Kit for Illumina (NEB #E7760) as recommended by the kit manufacturer and then sequenced with a Illumina NexSeq 550. RNA-seq samples were analyzed using the *nextpresso* pipeline[72] as follow: reads were aligned to the human genome (hg19) with TopHat-2.0.10[73] using Bowtie 1.0.0[74] and SAMtools 0.1.19[75], allowing 3 mismatches and 20 multi-hits; transcripts quantification was calculated with Cufflinks 2.2.1[73], using the human hg19 transcript annotations from https://ccb.jhu.edu/software/tophat/igenomes.shtml. Transcriptional subtypes were obtained using the 'ssgsea.GBM.classification' R package[76], through the SubtypeME tool of the GlioVis web portal (http://gliovis.bioinfo.cnio.es)[77].

**Mouse tumorspheres**. Mouse tumorspheres from ref. [56], deposited in GSE110700, were analyzed for NTRK transcript expression, as above, and subjected to immunoblotting and RT-qPCR.

*Immunoblotting.* Lysates were prepared in RIPA lysis buffer (20 mM Tris-HCl, 150 mM NaCl, 1% NP-40, 1 mM EDTA, 1 mM EGTA, 1% sodium deoxycholate, 0.1% SDS) and protein concentrations were determined by DC protein assay kit (Biorad, Cat. 5000111). Proteins were run on a 8% SDS-PAGE gel and transferred to nitrocellulose membrane (Amersham, Cat. GEHE10600003). After blocking the membrane in 5% milk, 0.1% Tween, 10 mM Tris at pH 7.6, 100 mM NaCl, primary antibodies TrkB (Merck, Cat. 07-225; lot: 3277578 at 1:3000) and vinculin (Sigma-Aldrich, Cat. V9131; lot: 118M4777V at 1:10.000) were added. Anti-mouse or rabbit-HRP conjugated antibodies (Jackson Immunoresearch) were used to detect desired protein by chemiluminescence with ECL (Amersham, RPN2106).

*Reverse transcription quantitative PCR* RNA was isolated with TRIzol reagent (Invitrogen, Cat. 15596-026) according to the manufacturer's instructions. For reverse transcription PCR (RT-PCR), 1 µg of total RNA was reverse transcribed using the High Capacity cDNA Reverse Transcription Kit (Applied Biosystems, Cat. 4368814). Quantitative PCR was performed using the SYBR-Select Master Mix (Applied Biosystems, Cat. 4472908) according to the manufacturer's instructions. qPCR was run using specific primers for TrkB.T1 (qmNtrk2T1Fw_CAGGTAG AACGGAGCAGCA and qmNtrk2T1RevA_GGTTAGCAGAGGGCAATGGA) and the TrkB.FL (qmNtrk2FLFwdA_TTCCTTGCCGAGTGCTACAA and qmNtrk2FLRevA_TCGTGCTGGAGGTTGGTC). The threshold cycle number for the genes analyzed was normalized to GAPDH (mGAPDHFwd_TCAACAGCA ACTCCCACTCTTCCA and mGAPDHRev_ ACCCTGTTGCTGTAGCCGT ATTCA).

**Generation of monoclonal antibody to TrkB.T1**. Murine monoclonal antibody (clone FH1D12) was generated at the Fred Hutchinson Antibody Technology Core Facility. Briefly, male and female 20-week-old C57BL6 mice that were characterized as genetic null for TrkB.T1[36] (gift from Francis S. Lee, Weill Cornell Medical College (New York, NY) and Lino Tessarollo, National Cancer Institute (Frederik, MD)) were immunized with TrkB.T1 peptide Ac-C(dPEG4)FVLFHKIPLDG-OH maleimide coupled to KLH carrier protein. Following a 12+ week boosting protocol splenocytes were isolated from high titer mice and electrofused to a subclone of the P3x63-Ag8 myeloma cell line (BTX, Harvard Apparatus). Hybridomas secreting peptide specific antibody were identified and isolated using a ClonePix2 (Molecular Devices, CPII) colony picker. Antibodies from the picked clones were validated for peptide binding by flow cytometry using a cytometric bead array carrying the target peptide. Clone FH1D12 was subcloned (CPII) followed by validation for peptide binding by bead-based flow cytometry. Affinity purified IgG2b lambda from the hybridoma was further characterized by Western blot analysis and IHC tissue staining. Antibody development was performed in Holland Lab and with the Antibody Technology Resource (Fred Hutchinson Cancer Research Center).

*Wes™ capillary electrophoresis for antibody development.* Protein Simple Wes™ was used at two stages of antibody screening: Stage 1, selecting immunized mice to advance for hybridoma generation, and Stage 2, for validation of monoclonal antibodies that Western blot full-length TrkB.T1 protein in murine brain detergent extracts. Briefly, mouse brain lysates from TrkB.T1[+/+] and TrkB.T1[−/−] mice were prepared by mechanical disruption (Dounce tissue homogenizer, Kimble Chase) in RIPA lysis buffer (Pierce™ RIPA Lysis and Extraction Buffer, ThermoScientific™)

containing HALT™ Protease and Phosphatase Inhibitor Cocktail and EDTA (ThermoScientific™) and normalized to 2 µg/µl total protein via Bradford Protein Assay. Approximately 8–10 ng total brain extract protein was loaded into the Protein Simple Capillary WES system for Western analysis following Protein Simple recommended methods[78]. Mice with antisera (1:2500 dilution in PBS and Antibody Diluent (Protein Simple)) that identified a positive band at 90–100 kDa for the TrkB.T1 protein were advanced forward for final boosting, splenectomy, and hybridoma generation (Supplementary Fig. 3). For Stage 2, antibodies from monoclonal culture super (diluted 1:2 in Antibody Diluent (Protein Simple)) targeting the TrkB.T1 peptide (see hybridoma generation and screening) were further tested for their ability to Western blot full length TrkB.T1 protein from normal and KO mouse brain detergent extracts. Clone FH1D12 (and additional clones—FH1D6 and FH1E6) was identified as having positive Western blotting activity (Supplementary Fig. 3) positive immunostaining in the wild-type (WT) mouse brain and human brain with absent staining in the TrkB.T1[−/−] knockout mouse brain.

*Recombinant expression of TrkB.T1 SPEH1_D12 scFv-Fc fusion protein.* The variable regions of the SPEH1_D12 (clone FH1D12) antibody were sequenced using rapid amplification of cDNA ends (RACE) cloning with standard primer sets and the SMARTer™ RACE 5′/3′ kit (Clonetech). The variable regions were then formatted into a fusion construct containing a rabbit antibody Fc domain (Supplementary Fig. 5). The amino acid sequence was then reverse translated using human codons, synthesized and cloned into a custom lentiviral expression vector. The construct was expressed as a soluble protein using the Daedalus expression method[79] and purified using a 5 mL MabSelect Sure column (GE)[79]. Following purification, the protein was supplemented with 5% glycerol, snap frozen using liquid nitrogen, and stored at −80 °C. Final stocks were 11.5 mg/ml in 85 mM Sodium Citrate 140 mM Tris-HCl with 5% glycerol.

**Immunohistochemistry**. *Antibody validation phase – large batch IHC.* Four-µm sections of a multi-tissue block containing human brain, mouse brain, TrkB.T1[−/−] mouse brain, a cell pellet over-expressing TrkB.T1 and a cell pellet without TrkB.T1 were cut and stained with the Leica Bond Rx (Leica Biosystems, Buffalo Grove, IL). Slides were pretreated with H2 antigen retrieval buffer for 20 min. Endogenous peroxidase was blocked with 3% hydrogen peroxide for 5 min. A TCT protein block was applied for 10 min (0.05 M Tris, 0.15 M NaCl, 0.25% Casein, 0.1% Tween 20, pH 7.6). Supernatants from multiple boosting and pre-/post-fusion samples, along with antibody clone D12 were used at 1:20 and applied to the tissue for 30 min. The antibody was then detected using Leica Power Vision HRP Mouse specific polymer (Leica Biosystems catalog# PV6114) for 12 min and staining was visualized with Refine DAB (Leica Biosystems catalog# D59800) and a hematoxylin counterstain was used (Hematoxylin 50% in H2O from Biocare catalog #NM-HEM-M). Mouse isotype control slides were included for each run (Jackson ImmunoResearch Laboratories) at 2 µg/ml. Digital images of IHC-stained slides were obtained at ×40 magnification (0.25 µm/pixel) using a whole slide scanner (ScanScope AT Turbo, Aperio) fitted with a 20×/0.75 Plan Apo objective lens and ×2 magnification changer (Olympus, Center Valley, PA, USA). Images were saved in SVS format (Aperio), managed with server software (ImageServer, Aperio), and retrieved with a file management web interface (eSlideManager, Aperio). Histology and IHC for antibody validation was performed by Experimental Histopathology Shared Resources at Fred Hutchinson Cancer Research Center.

*Mouse brain, mouse glioma, human brain,* and *human glioma.* Immunohistochemical staining was performed on 5um formalin-fixed/paraffin-embedded tissue sections using a Discovery XT Ventana Automated Stainer (Ventana Medical Systems, Inc)., run using standard Ventana reagents and the Discovery ChromoDAB and Discovery OmniMAP anti-ms HRP kits to ameliorate non-specific mouse-on-mouse background and standard Ventana reagents and Vector secondaries for staining with the TrkB.T1 SPEH1_D12 scFv-Fc fusion.

**Human tissue and case selection**. Approval for the use of human subject material was granted by the University of Washington's Institutional Review Board (Study #44806 and #00002162). Archived FFPE pathology tissue were reviewed by a board-certified neuropathologist (P.J.C.) and classified according to the 2016 WHO classification of CNS tumors: Oligodendroglioma, IDH-mutant and 1p/19q-code-leted (n = 5); Astrocytoma, IDH-mutant (n = 5); and Glioblastoma, IDH-WT (n = 14 total; n = 5 classical, n = 4 proneural, n = 5 mesenchymal)[37,80]. Normal brain control tissue from rapid autopsies (n = 5) was procured from the University of Washington's Neuropathology Core Brain Aging and Neurodegeneration Brain Bank. Each control case contained multiple brain regions, including cerebral cortex, cerebellum, hippocampus, subiculum, entorhinal cortex, and thalamus.

**Mouse tissue processing**. Mouse tissue (including normal brains, tumor bearing brains) were removed, fixed in 10% neutral-buffered formalin for a minimum of 72 h and embedded into paraffin blocks. 5 µm serial sections were cut from formalin-fixed paraffin-embedded specimens and mounted on slides.

**NSC experiments**. *Cell culture.* GSC and NSC isolates were grown in NeuroCult NS-A basal medium (StemCell Technologies) supplemented with B27 (Thermo-Fisher), N2 (2× stock in Advanced DMEM/F-12 (ThermoFisher) containing

25 µg/mL insulin, 100 µg/mL apo-transferrin, 6 ng/mL progesterone, 16 µg/mL putrescine, 30 nM sodium selenite, and 50 µg/mL bovine serum albumin (Sigma)), and EGF and FGF-2 (20 ng/ml) (PeproTech) on laminin (Trevigen or in-house-purified)-coated polystyrene plates and passaged[81]. Cells were detached from their plates using Accutase (EMD Millipore).

*Lentiviral production and infection.* For virus production, pLJM1 (Addgene) constructs containing the NTRK2 inserts of interest (TrkB.FL, TrkB.T1, or GFP) were transfected into 293T cells, along with psPAX and pMD2.G packaging plasmids (Addgene), using polyethylenimine (Polysciences). Fresh media was added 24 h later and viral supernatant harvested 24 h after that. For infection of NSC-U5 cells, 1e5 cells/well were seeded into 6-well plates. Lentivirus was used unconcentrated and cells were infected at a MOI < 1 24 h after seeding. 72 h after seeding, selection was begun for cells successfully expressing the constructs using 2 µg/mL puromycin (for 3 days). Cells were expanded after selection to create a stable line and collected for RNAseq three weeks post infection.

*PI3K and mTOR inhibitors.* In order to assess the sensitivity of TrkB.T1 and TrkB.FL overexpressing NSCs to LY294002 and rapamycin, we used biological triplicates of each line and seeded $6 \times 4000$ cells per triplicate into four laminin coated 96-well plates. One day after seeding, LY294002 (Life Technologies™ #PHZ1144; Lot #76075413) and rapamycin (Life Technologies™ #PHZ1235; Lot #2142418) were dissolved in DMSO warmed to 37 degrees at 10 mM, diluted to 10 µM in NSC media and added to cells in 100 µl NSC media for a total volume of 200 µl at specified concentrations. After 48 and 72 h of drug treatment, plates were analyzed using *CellTiter-Glo® Luminescent Cell viability Assay* (CTG, Promega). In brief: CTG reagent was reconstituted according to the manufacturer's instructions. The reagent was then diluted 1:5 in PBS. The 96-well plates were equilibrated to room temperature and existing NSC media with drugs was removed before 100 µl of diluted CTG reagent was added to each well. The cells in the plates were then lysed for 2 min with shaking, followed by a 10 min incubation at room temperature. 90 µl of the cell lysate in each well was transferred to opaque 96-well plates for detection. The emitted ATP-driven luminescence signal was detected using the Synergy 2 instrument (BioTek) and the Software Gen5. Integration time was set at 1 s and sensitivity to 135.

**RCAS-TrkB.T1**. Flag-tagged TrkB.T1 and Flag-tagged TrkB.FL DNA were cut from pEFBOS vectors (generous gift from Eero Castren[23]), visualized on a 1% agarose gel and extracted via QIAquick gel extraction kit (Qiagen), and ligated into modified RCAS (replication-competent avian sarcoma-leukosis virus (ASLV) long-terminal repeat with splice acceptor) vectors[82]. Appropriate inserts were confirmed by sequencing at Fred Hutch Genomics Core. RCAS virus was produced in DF-1 packaging cells with minor modification[83,84]. In brief, DF-1s (ATCC) were maintained in Dulbecco's Modified Eagle Media (DMEM) with 1% penicillin/streptomycin and 10% fetal bovine serum (Paa Laboratories) at 39 °C, were transfected with RCAS-TrkB.T1 using X-tremeGENE9 (Roche) + Opti-MEM and passaged three times at high confluency to enhance viral propagation. Expression for TrkB.T1 was confirmed by western blot after three passages while TrkB.FL was at maximum size limitations for RCAS and did not make TrkB.T1 protein.

**Generation of murine tumors**. We utilized the RCAS/tv-a system for murine tumor modeling in immunocompetent mice[83]. In brief, DF-1 cells were transfected with the relevant RCAS viral plasmids (RCAS-PDGFB or RCAS-TrkB.T1) using Extreme-Gene 9Transfection reagent (Roche) accordingly to manufacturer's protocol. The cells were maintained for three passages to ensure viral propagation to all cells. After confirmation of RCAS-inserts by western blot, DF1s (passage 4 or later) were used for injection into murine brain. Newborn *Nestin(N)/tv-a*(agouti) pups (P0-P1; males and females) were injected intracranially (Hamilton syringe #84877) with 1 µL of approximately $1 \times 10^5$ DF-1 cells infected with and producing relevant RCAS viruses suspended in serum-free DMEM (RCAS-TrkB.T1 and RCAS-PDGFB). Simultaneous delivery of two RCAS viruses was performed by the injection of 1 µL of ~$2 \times 10^5$ DF-1 cells mixed with equal ratio. Mice were monitored for weeks or months to check for tumor related symptoms such as palpable masses, lethargy, weight loss, seizure, hyperactivity, altered gait, poor grooming, macrocephaly, paralysis. Mice with severe hydrocephalus presumably due to injection trauma or an inflammatory response against the DF-1 cells were excluded from survival analysis in this study. All animal experiments were approved by and conducted in accordance with the Institutional Animal Care and Use Committee of Fred Hutchinson Cancer Research Center (protocol #50842).

**3T3/tv-a cells**. Stable NIH-3T3 cells expressing *Tv-a* receptor[39] were subjected to retroviral infection with RCAS-TrkB.T1 via $3 \times 8$ h cycles using sterile filtered, viral-containing supernatant from DF-1 cells supplemented with 8 µg/ml polybrene (Sigma) and later selected with 2.5 µg/ml puromycin (GIBCO). 3T3/tv-a cells were maintained for three passages prior to any experiments and cultured in DMEM containing penicillin/streptomycin and 10% fetal calf serum (FCS; newborn calf serum filtered, heat-inactivated ThermoScientific). Prior to PDGF treatment, 3T3-tva cells were serum-starved for 24 h. Recombinant human PDGF-BB (Peprotech, catalog# 1001-14B, lot #071504) was diluted in fresh serum-free DMEM and added for various time points before cells were lysed on ice in RIPA extraction buffer

(Pierce™ RIPA Lysis and Extraction Buffer containing HALT™ Protease and Phosphatase Inhibitor Cocktail; ThermoScientific™) after being rinsed twice with Dulbecco's phosphate buffered saline (DPBS)(Gibco, -Ca²⁺ and -Mg²⁺). Extracts were further lysed by mechanical disruption using 18 ½ gauge needles, rotated at 4 °C for 20 min to ensure lysis, and clarified by centrifugation ($16,000 \times g$ for 10 min at 4 °C). Proteins were resolved by SDS/PAGE (NuPAGE 10% Bis/Tris; LifeTech) according to XCell Sure Lock™ Mini-Cell guidelines, blocked with 5% milk/TBST and probed with specified antibodies overnight at 4 °C in 5% BSA/TBST. After three TBST rinses, species-specific secondary antibodies were added in 5% milk/TBST at 1:10,000. Blots were rinsed three times with TBST before being developed with Amersham™ ECL™ Western Blotting Detection Reagents (GE Healthcare). For densitometric analyses, immunoreactive bands were scanned and quantitated using National Institutes of Health ImageJ (Scion).

**Antibodies**. For TrkB.T1 IHC, the generated mouse TrkB.T1 antibody was used (1.3 mg/ml) at 1:20 and recombinant rabbit fusion SPEH1_D12 (11.5 mg/ml) was used at 1:500 and TrkB kinase antibody (abcam #ab18987; lot: GR3280550-2) at 1:250. For western blot analyses of human brain, mouse brain, and 3T3-Tv-a cells, commercial antibodies were used according to manufacturer specifications and bands were confirmed by size using Spectra™ Multicolor Broad Range Protein Ladder (ThermoScientific™ catalog # 26634; lot: 00784968): pSTAT3 (Tyr705, Cell Signaling #9145; lot: 22 at 1:1000), pAkt (Ser473, Cell Signaling #4060; lot: 23 at 1:1000), pERK (Cell Signaling phospho-p44/42 MAPK (ERK1/2) (Thr202/Tyr204) (D13.14.4E) XP® #4370; lot: 12 at 1:2000), β-actin (Sigma #AB1978 Clone AC-15; lot: 021M4821 at 1:10,000), pS6 ribosomal protein (Ser235/236, Cell Signaling #4858; lot: 16 at 1:1000), TrkB (Millipore #07-225; lot: 2187222 and lot: 3277578 @ 1:3000), TrkB (abcam #ab33655; lot: GR266297-1 at 1:1000), TrkB (Abcam #ab18987; lot: GR3280550-2 at 1:2000), pPDGFR β (Y1021; Abcam #ab62437; lot: GR38791-13 at 1:200), PDGFR β (Cell Signaling #3169; lot: 13 at 1:800), ERBB3 (Cell Signaling #12708; lot: 4 at 1:1000), NRG2 (Abcam #ab220615; lot: GR3181158-4 at 1:200), vinculin (Sigma-Aldrich, Cat. V9131; lot: 118M4777V at 1:10.000) and added in 5% BSA/TBST overnight at 4 °C.

**Primary *Nestin/tv-a* neurospheres**. Neural progenitors were isolated from *Nestin(N)/tv-a*pups at P0. A minimum of four P0 *N/tv-a* WT pups were used. SVZ/Hippocampal-centric region was dissected and placed immediately in 3 ml of accutase warmed to 37 °C. Cells were pipetted 30 times for mechanical dissociations and left in a 37 °C water bath with occasional agitation before being pipetted an additional 30 times with a P1000 to create a single cell suspension. Two additional mLs of NSC media were added, cells centrifuged @ $200 \times g$ for 3 min at 4 °C. Supernatant was removed and pellet was resuspended in 5 ml NSC media, transferred to a new tube (leaving any debris behind) and seeded 100 µL into 4 ml NSC media in 25 cm² flask. 50 ml fresh NSC media were made each day: 50 ml NSC media consisted of (45 ml Neurocult Mouse Basal media, 5 ml Neurocult Proliferation Supplement (mouse), 50 µL EGF (20 µg/ml stock), 50 µL bFGF (10 µg/ml stock), 125 µL Heparin (Stem Cell Technologies #07980), 500 µL P/S). Fresh growth factors were added every three days. Neural progenitors were passaged twice prior to the start of experiments. For retroviral infection of murine neurospheres with RCAS-TrkB.T1 or RCAS-GFP, RCAS virus was produced in DF-1 packaging cells maintained with serum-free neurosphere medium and was then diluted 1:1 with collection media (fresh murine neurosphere media). Efficiency of retroviral infection was confirmed via western blot after one passage and images were obtained in Fred Hutch Imaging Core.

**Reporting summary**. Further information on research design is available in the Nature Research Reporting Summary linked to this article.

## Data availability
RNASeq data that support the findings of this study have been deposited in GEO database under the accession code GSE136868. Bulk gene expression data for Mouse tumorspheres was obtained from GSE110700. Bulk gene expression data for cell line 448, 559, G14, and G179 was obtained from GEO repository, GSE89623. Bulk gene expression counts for 44 Human GSCs was obtained from GSE119834. Bulk gene expression counts for six additional Human GSCs (BTSC349, BTSC349, h543, h516, h561 and h676) was obtained from GSE150653. RNA-seq data for human normal brain samples were downloaded from the GTEx data portal (v6) [https://www.gtexportal.org/]. RNA-seq data for TCGA-GBM and TCGA-LGG were downloaded from the UCSC Xena/Toil hub [https://xenabrowser.net/datapages/?hub=https://tcga.xenahubs.net:443]. Transcript data for TCGA-GBM and TCGA-LGG were downloaded from Broad's firebrowse [http://firebrowse.org/?cohort=GBM&download_dialog=true]. All other relevant data are available from the corresponding author on request. The data that support the findings of this study are included with the manuscript and supplementary data files and also available from the corresponding author upon reasonable request.

## Code availability
All custom scripts have been made available at https://github.com/sonali-bioc/Pattwelletal2020. All analyses were performed using publicly available software (R and R/Bioconductor packages) as indicated in Materials and Methods.

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

## Acknowledgements

We thank James Yan, Jenny Zhang, Deby Kumasaka, and Denis Adair for continued technical and administrative assistance and support throughout these experiments. Eero Castren at University of Helsinki provided pEF-BOS-TrkB plasmids used for RCAS-TrkB.T1 generation. We thank Francis S. Lee and Lino Tessarollo for providing breeding pairs of TrkB.T1$^{-/-}$ mice for antibody production. Luis Chiriboga at New York University School of Medicine provided helpful insight into histology protocols. We thank William A. Johnsen and Midori Clarke for assistance with antibody sequencing and protein purification. We thank the Tracy A. Goodpaster and Julie Randolph-Habecker at the Fred Hutchinson Experimental Histopathology Core for histology assistance during the antibody validation phase and Elizabeth Jensen at the Fred Hutchinson Genomics Core for help with all DNA sequencing. We thank Jeongwu Lee Do-Hyun Nam and Steven M. Pollard for providing cell isolates. Funding was provided by National Institutes of Health R01 CA195718 (E.C.H.), U54 CA193461 (E.C.H., F.S.), R01 CA100688 (E.C.H.), T32 CA965725 (S.S.P.), U54 DK106829 (S.S. P.), R21 CA223531 (S.S.P., E.C.H.), T32 CA080416 (P.H.), R01 CA190957 (P.J.P.; T. B.); Jacobs Foundation Research Fellowship (S.S.P.); American Cancer Society ACS-RSG-14-056-01 (P.J.P.); National Research Agency RTI2018-102035-B-I00 (M.S.); Seve Ballesteros Foundation (M.S.). Autopsy materials used in this study were obtained from the University of Washington Neuropathology Core, which is supported by the Alzheimer's Disease Research Center (AG05136), the Adult Changes in Thought Study (AG006781), and Morris K Udall Center of Excellence for Parkinson's Disease Research (NS062684)

## Author contributions

S.S.P. proposed the scientific concepts, designed and performed experiments, generated data, assisted with antibody validation and bioinformatic analyses, and wrote the manuscript. S.A. and H.B. performed bioinformatic analyses, designed bioinformatic pipelines, and generated data. P.J.C. examined all tissue slides for human and mouse, generated photomicrographs, offered neuropathology expertise and contributed to the manuscript. J.S. provided tissue for background studies and antibody validation. T.O. provided necessary reagents, supervised initial experimental design, and offered technical guidance throughout. F.S., P.H., and T.B. were instrumental with cloning assistance, lentiviral design, and NSC experiments. P.J.P. provided cell lines and data for NSC and GSC experiments. B.O. and M.S. provided and analyzed data from mouse tumorsphere and human GSC lines. B.G.H., N.E.B., and C.E.C. worked on antibody generation and validation. J.R.S. collected and annotated human glioma samples. E.C.H. contributed to the overall experimental design, supervised the project, and offered critical feedback throughout the project and manuscript revisions. All authors discussed data and contributed to the final manuscript.

## Competing interests

The authors declare no competing interests.
