## [Peer Review File · Nature Communications]

Reviewers' Comments:

Reviewer #1:

Remarks to the Author:

Pattwell and colleagues demonstrated that a TrkB splice variant, TrkB.T1, is highly expressed in glioma compared to normal brain tissue. Their novel antibody clearly showed the distribution of TrkB.T1 in glioma cells. Additionally, TrkB.T1 enhanced gliomagenesis in their RCAS/tv-a system. TrkB.T1 augmented PDGF-induced Akt and STAT3 signaling in vitro and RNAseq of GFP, TrkB1.T1, and TrkB1.FL overexpressing NPCs implicated ligand-independent function of TrkB.T1 in PI3K/Akt and PI3K/ERBB2 signaling cascades. Overall the data are very clear and conclusions justified, however there are several issues that need to be addressed as detailed below, which will enhance the overall significance of TrkB.T1 in glioma.

Major

1. Survival data in Supplementary Fig.1 needs to be shown for LGG and GBM separately and clarify if TrkB.T1 is related to GBM patient survival.
2. The authors have shown exogenous TrkB.T1 enhances tumorigenesis in Fig.5, however it is not shown how this gene enhances tumor survival and resistance to standard therapy. Treatment experiments should be done at least in vitro to implicate a clinical role for this gene.
3. Does TrkB.T1 affect TrkB1.FL function? Do they heterodimerize? If the RCAS model in Fig. 5A was knocked down for TrkB1.FL, does TrkB.T1 still enhance tumorigenicity?
4. It is difficult to make conclusions based on one high and one low TrkB.T1-expressing GSCs in Fig 6B. At a minimum, 2 human GSC cell lines for low and high should be analyzed.
5. Findings for key signaling effectors in Fig7 need to be validated by western blot.

Reviewer #2:

Remarks to the Author:

Major concerns:

- 1) Western blot shown in Figure 2C is insufficient. Needs to be repeated, include more normal brain samples, more glioma samples. Uncropped blots and lighter exposure must be shown in supplement. Also, RNA expression data from GTEx and TCGA datasets shows equal levels of full length and short transcripts in normal brain, but Western blot makes it look like much more abundant short protein isoform is present in the normal brain (as well as gliomas). Authors should comment on this. Is there difference in protein stability between full-length and short isoforms?
- 2) The data in Figure 3 seem biologically meaningless to me. The authors do not show any functional assessment linking TrkB splicing to these differences in gene expression.
- 3) Figure 4 – data proving specificity of the newly developed antibody for IHC on FFPE tissue is lacking. In fact, the data shown actually prove the antibody lacks any specificity whatsoever, as the same pattern of vesicular cytoplasmic staining is shown in both normal (TrkB wildtype +/-) mouse cortex as well as TrkB.T1-/- cortex. What's the deal with this?
- 4) The data in Figure 6 seem biologically meaningless to me. Correlating gene expression levels

between a glioma cell line from one patient that happens to be TrkB.T1 low and another patient that happens to be TrkB.T1 high is merely correlative mumbo jumbo and is overall meaningless.

5) The analysis in Figure 7 attempts to show a direct effect between TrkB isoforms and gene expression via RNA-seq after infection in a neural stem cell line. However, the analysis seems very weak overall and the conclusions are blown way out of proportion for a such a limited study in a single cell line without meaningful biological replicates. Also it is unclear why NTRK2 transcript abundance appears to be increased only after infection with TrkB.T1 and not TrkB.FL. What's the deal with this? Doesn't this call into question the validity of the entire figure?

Reviewer #3:

Remarks to the Author:

In the manuscript entitled A kinase-deficient NTRK2 splice variant predominates in glioma and amplifies several oncogenic signaling pathways, Dr. Pattwell and colleagues evaluate the role of the TRKB splicing variant TrkB.T1 in human and mouse brain tumors with focus on glioblastoma (GBM) and Lower grade Glioma (LGG).

Given the important role of neurotrophins and their receptors (TRKA/TRKAB/and TRKC) in the development of the nervous system and the fact that fusions involving genes encoding the TRK receptors have been found as recurrent genetic drivers in brain tumors, the manuscript focuses on a topic of extreme scientific interest and clinical relevance.

However, I have several major concerns regarding the approach taken as well as the experimental validation of the authors' observations that I personally do not think always support their conclusions.

Major points

1) As an initial broad analysis, the authors analyze the expression levels of the whole gene NTRK2 across normal brain, LGG and GMB using two distinct datasets [GTEx and TCGA (2 studies)]. The expression of the whole gene was unchanged across normal brain, LGG and GBM. Nevertheless, the authors speculate that a differential expression of the different splicing variants of this gene may exist and compare two of them, the TrkB-FL and the kinase-deficient TrkB.T1. This is a biased choice that is just justified by the fact that these two isoforms are the most studied. To give a stronger rationale on why the authors further focus on the TrkB.T1 isoform, the analysis should have included at least all the splicing variants of TrkB (if not all the variants of all NTRK genes) (PMID: 30333516). If TrkB.T1 confirms to be the most highly differentially isoform, the rationale to focus on this variant would be much stronger than it is right now.

2) The authors use WB analysis to show that TrkB.T1 expression at protein level is higher in glioma when compared to normal human cortex (fig. 2C). They discriminate between the FL and the T1 forms based on the size. There is overlay between the two band in sample 2 (glioma), thus rendering very hard to understand the differences in levels compared to the normal. Moreover, this is just 1 sample vs 1 control. This WB is not informative and should be removed.

3) The authors realize the need of a better tool to discriminate the different isoforms and design a T1-specific antibody that recognize the unique C-term of this variant. They validate its specificity in different ways using tissues collected from mice with different TrkB-T1 background. The validation here is crucial but Supplementary figure 3 is very blurry. Also, I think that now the authors should present the WB (Fig 2C), with the normal antibody as well the TrkB.T1 specific. In this case just 1 band in both samples should show up with the T1-specific and maybe the differential abundance among the normal and the tumor can be better appreciated.

- 4) The authors stained multiple glioma models with different background (EGFR, PDGF, PTEN null, NF1 null) with the T1 antibody and found strong, similar positivity in all samples. However, for all the following experiments, they focused on a PDGF-driven model? Which is the rationale? EGFR signaling is often altered in glioma and, as shown in the results presented in Figure 3D (GO analysis), EGFR signaling is a hit as a differentially expressed pathway between tumor and normal. This is again a biased choice that should be avoided or, alternatively, other hits should be studied further for comparison. For example, transduced (with FL and T1) 3T3 and NSCc should be tested following EGF and BDNF (as a control) stimulation. Also, the authors suggest a cross-talk between T1 and RTKs (overall PDGF). CO-IPs should be performed to test whether they bind.
- 5) Data presented in Figure 6 on the 448T and 559T cells are not supportive of the authors' conclusions. These cell lines were chosen for their differential expression of T1 and compared at the level of transcriptome. The fact that two different cell lines have differential gene expression is absolutely normal and the authors can not conclude that these differences are due to the different expression of T1. If the authors want to claim this, they should, for example, knock-down T1 (maybe targeting the unique C-term sequences) in the high T1-expressor 448T, check the KD with their new antibody and then run RNA seq on this pair. The same can be done using the other cell line following T1 overexpression. If these 2 experiments are not performed the whole section is inconclusive and needs to be removed.
- 6) Following the final transcriptomic analysis on the NSCs transduced with FL and T1, the authors conclude that the PDGF/PI3K/AKT/mTOR axis is activated preferentially in the presence of T1. No validation is presented. Moreover, if this is correct, these cells should respond better to PDGF/AKT/PI3K/mTOR inhibitors when compared to the FL transduced cells. These experiments as proliferation assays and biochemical WB-based assays should be performed.
- 7) Figure 7C the expression of TRKB in the T1 transduced cell is much higher than in cells transduced with TrkB FL. The WB showing the transduced cell line is required since these cells are compared

Minor points:

- 1) On page 3, when "several, lesser known, alternative spliced variants" are mentioned, the statement should be referenced (For example PMID: 30333516)
- 2) I am not sure Fig 1b should be a main one. It is already clear from the 1A that the rest of the samples (once cerebellum and spinal cord data are removed) cluster together. I would move to the Suppl
- 3) Please indicate statistic for the survival curve presented in suppl Fig 1. Also, I think this is an interesting point that the authors may want to highlight as it may have strong clinical implications (T1 could be a biomarker of prognosis tested with the T1-specific antibody)
- 4) I am not sure Suppl Fig 4C have the images in the same magnification when compared with panel A and B. Please change for consistency
- 5) In the WB conducted on the transduced 3T3 the authors should show pPDGF, total PDGF, pERK and ERK
- 6) Figure 4 the authors show no staining with T1 antibody in the normal. It would be nice to see the staining for the FL
- 7) In the discussion "the data shown here reveal a kinase-deficient isoform, T1, to be the predominant isoform in brain tumors compared to normal brain". This is an overstatement since the authors just checked the expression of this variant
- 8) Pag 15 "and future studies should explore if this IS a PDGF....."

NCOMMS-19-34614-T
Point-by-point Response to Reviewers

We thank the Reviewers for the insightful and constructive suggestions to our manuscript. We have addressed their comments and requests in the following point-by-point summary. In addition, we have added several recent citations throughout the manuscript, where appropriate, that were not yet available at the time of initial submission. We have also clarified headings, descriptions, and schematics in our main and supplementary figures to be inclusive of any initial and newly added data and thank the Reviewers for strengthening our manuscript with their comments and suggestions.

Reviewers' comments:

Reviewer #1 (Remarks to the Author):

Pattwell and colleagues demonstrated that a TrkB splice variant, TrkB.T1, is highly expressed in glioma compared to normal brain tissue. Their novel antibody clearly showed the distribution of TrkB.T1 in glioma cells. Additionally, TrkB.T1 enhanced gliomagenesis in their RCAS/tv-a system. TrkB.T1 augmented PDGF-induced Akt and STAT3 signaling in vitro and RNAseq of GFP, TrkB1.T1, and TrkB1.FL overexpressing NPCs implicated ligand-independent function of TrkB.T1 in PI3K/Akt and PI3K/ERBB2 signaling cascades. Overall the data are very clear and conclusions justified, however there are several issues that need to be addressed as detailed below, which will enhance the overall significance of TrkB.T1 in glioma.

Major

1. Survival data in Supplementary Fig.1 needs to be shown for LGG and GBM separately and clarify if TrkB.T1 is related to GBM patient survival.

We thank the Reviewer for this suggestion. We agree the survival data should be separated into LGG and GBM and presented it in this way mainly to clarify a discrepancy in a recent paper (Deluche et al., 2019, *Cancers*, "CHI3L1, NTRK2, 1p/19q and IDH Status Predicts Prognosis in Glioma), in which the authors did not separate survival curves by brain tumor type and have a typographical error in their published manuscript, where one of the figures (4a-d) shows a red line (low expression) and a blue line (high expression) for which the HIGH expression of NTRK2 (blue line) appears to have worse prognosis while the LOW expression of CHI3L1 (red line) appears to have worse prognosis. Yet, the text of the same page, says the opposite: "In our cohort (Figure 4a,c), and the glioma TCGA cohort (Figure 4b,d), low expression of NTRK2 and high expression of CHI3L1 were strongly linked to poor prognosis ($p < 0.05$)."

We have maintained our initial data as a way to clarify this published discrepancy for the field to avoid confusion and also separated the curves accordingly as per Reviewer #1's suggestion and additionally stratified LGG and GBM by CIMP and non-CIMP status. These data demonstrate that high expression of the NTRK2 kinase does *not* lead to worse survival in either LGG or GBM, as previously suspected by others, and is now incorporated into Supplementary Fig. 1a-e.

2. The authors have shown exogenous TrkB.T1 enhances tumorigenesis in Fig.5, however it is not shown how this gene enhances tumor survival and resistance to standard therapy. Treatment experiments should be done at least in vitro to implicate a clinical role for this gene.

We thank the Reviewer for this suggestion and have treated human neural stem cells (NSCs) transduced with either TrkB.FL or TrkB.T1 with varying concentrations of PI3K inhibitor (LY294002) or mTOR inhibitor (rapamycin) and tested for proliferation differences and viability via CellTiter-Glo assay which determines the number of viable cells in a culture based on quantification of ATP, an indicator of metabolically active cells. As these NSCs are stem-like by nature, both drugs worked in a dose dependent manner to reduce viability and TrkB.T1-transduced NSCs were marginally, but statistically significantly, more sensitive to LY294002, which is in line with data throughout the manuscript suggesting a role for TrkB.T1 in modulating PI3K signaling.

This is shown in our revised manuscript:

“Treatment of NSCs transduced with TrkB.FL and TrkB.T1 showed that both lines were sensitive to phosphoinositide-3 kinase (PI3K) and mTOR inhibitors (LY294002 and rapamycin, respectively) in a dose dependent manner and that TrkB.T1 NSCs were marginally, but statistically significantly, more sensitive to LY294002 than TrkB.FL NSCs at both 48 and 72 hours (Supplementary Fig. 8d, 8e). Combined with RNASeq data suggesting a role for TrkB.T1 in PI3K signaling and enhancement of downstream PDGFR pathways *in vitro* (Fig. 5, Fig. 7, Supplementary Fig. 8, Supplementary Datasheet 6), increased sensitivity to LY294002 in the TrkB.T1 NSCs suggests that TrkB.T1 may selectively activate PI3K pathways.”

We also explored our RNASeq data for a proliferation index (as previously described in newly added references Ramaker et al. (2017) *Oncotarget* (8)24: 38668-38681 and Venet et al. (2011) *PLoS Comput Biol*(7)10: e1002240). As these NSCs are stem-like by nature, proliferation indices are high overall due to the cell type and culture conditions required for stemness, however, TrkB.T1 transduced NSCs do show higher expression of genes in this proliferation index compared to TrkB.FL as shown in Supplementary Fig. 9.

3. Does TrkB.T1 affect TrkB1.FL function? Do they heterodimerize? If the RCAS model in Fig. 5A was knocked down for TrkB1.FL, does TrkB.T1 still enhance tumorigenicity?

We thank the Reviewer for this insightful comment. While we are planning to perform *in vivo* experiments (using RCAS-PDGFB+shTrkB.FL +/- RCAS-TrkB.T1), they are currently outside the scope of this manuscript and additional rodent RCAS studies fall outside of the allowable time frame for resubmission. We do feel confident, based on our *in vitro* data, that this PDGF-driven effect observed in our mouse model is *not* driven by TrkB.T1-TrkB.FL heterodimers because 3T3 cells do not express TrkB.FL and therefore the observed pAkt, pSTAT3, pS6 effects could not be dependent on TrkB.FL. We have emphasized this in our revised text and it can be shown in Fig. 5b which now highlights the position of TrkB.FL (~145kDa) and TrkB.T1 (~90kDa) in the western blot, compared to mouse brain as a control. We have emphasized this in our revised text via a newly added reference (Fryer, R.H., Kaplan, D.R. & Kromer, L.F. *Exp Neurol* **148**, 616-27 (1997)) and as can be shown in our revised Fig. 5b.

4. It is difficult to make conclusions based on one high and one low TrkB.T1-expressing GSCs in Fig 6B. At a minimum, 2 human GSC cell lines for low and high should be analyzed.

We thank the Reviewer for this suggestion and have included two additional GSC cell lines for high and low TrkB.T1 expression. To ensure we were not biasing our data, we chose GSC human cell lines derived from distinct tumor subtypes, in this case, with a mesenchymal phenotype (G14 and G179), to compare to our initial proneural/classical lines (448T and 559T). We have included this new data in the revised manuscript, comparing the different GSC lines to each other, and also the high TrkB.T1 expressing lines to the low TrkB.T1 expressing lines, and believe this addition of not only adding two additional lines, but also two distinct *types* of GSCs (mesenchymal and proneural) strengthens our GSC approach for not only this manuscript, but for future investigation in these cell lines. We have also maintained this subtype level analysis in newly added analysis of 50+ additional human GSC lines, shown in Fig. 2 and Supplementary Fig. 2.

5. Findings for key signaling effectors in Fig7 need to be validated by western blot.

We have included western blots for several effectors in Supplementary Fig. 9, including for TrkB itself, as suggested by other Reviewers.

Reviewer #2 (Remarks to the Author):

Major concerns:

1) Western blot shown in Figure 2C is insufficient. Needs to be repeated, include more normal brain samples, more glioma samples. Uncropped blots and lighter exposure must be shown in supplement. Also, RNA expression data from GTEx and TCGA datasets shows equal levels of full length and short transcripts in normal brain, but Western blot makes it look like much more abundant short protein isoform is present in the normal brain (as well as gliomas). Authors should comment on this. Is there difference in protein stability between full-length and short isoforms?

As requested by Reviewer #3, we have removed this western blot from the manuscript. We have added additional data, as shown in Fig. 2, Supplementary Fig. 2, and Supplementary Datasheets 2 and 3, showing TrkB.T1 to be the predominant NTRK isoform across a large selection (50+) of human GSC and rodent tumorsphere lines (Supplementary Fig. 7, Supplementary Datasheets 2 and 3) via transcript analyses, western blot, and qPCR.

2) The data in Figure 3 seem biologically meaningless to me. The authors do not show any functional assessment linking TrkB splicing to these differences in gene expression.

We apologize for any confusion and/or if our claims were misleading in terms of implying a cause and effect nature in Fig. 3. The analysis performed, DGCA, or Differential Gene Correlation Analysis is not, and was not intended to imply causality. Simply, DGCA is a package for dissecting regulatory relationships between genes in distinct conditions and has been used successfully to identify changes in regulatory relationships between genes (such as TP53 and PTEN as well as their target genes) in TCGA breast cancer samples. We performed this analysis as a correlational guide to potentially narrow down the list of interactors to inform our subsequent experiments, such as the use of PDGFB, etc., which we then proceeded to follow up with for *in vitro* and *in vivo* studies.

3) Figure 4 – data proving specificity of the newly developed antibody for IHC on FFPE tissue is lacking. In fact, the data shown actually prove the antibody lacks any specificity whatsoever, as the same pattern of vesicular cytoplasmic staining is shown in both normal (TrkB wildtype +/-) mouse cortex as well as TrkB.T1^{-/-} cortex. What's the deal with this?

We thank the Reviewer for raising this point and apologize for any confusion for interpretation of our figures. We have included additional IHC images to clarify that we do not see any staining whatsoever in the TrkB.T1^{-/-} cortex, which is in contrast to the punctate, vesicular pattern in wild-type mouse cortex (and also human cortex). Additionally, we have included IHC for the full-length isoform, TrkB.FL, using an antibody designed against the kinase of this receptor (specifically amino acid ~810 of TrkB) and it very clearly shows a pattern distinct from that of TrkB.T1. Unlike the pattern observed with TrkB.T1, TrkB.FL is present in the TrkB.T1^{-/-} cortex, present in WT cortex, but absent

in tumors. These collective images are found in Supplementary Fig 5 and Supplementary Fig. 6.

4) The data in Figure 6 seem biologically meaningless to me. Correlating gene expression levels between a glioma cell line from one patient that happens to be TrkB.T1 low and another patient that happens to be TrkB.T1 high is merely correlative mumbo jumbo and is overall meaningless.

In accordance with Reviewer #2's suggestion of adding an additional pair of GSCs, we have included two additional GSC cell lines for high and low TrkB.T1 expression. We thank the Reviewer for this suggestion and have included two additional GSC cell lines for high and low TrkB.T1 expression. To ensure we were not biasing our data, we chose GSC human cell lines derived from distinct tumor subtypes, in this case, with a mesenchymal phenotype (G14 and G179), to compare to our initial proneural/classical lines (448T and 559T). We have included this new data in the revised manuscript, comparing the different GSC lines to each other, and also the high TrkB.T1 expressing lines to the low TrkB.T1 expressing lines, and believe this addition of not only adding two additional lines, but also two distinct *types* of GSCs (mesenchymal and proneural) strengthens our GSC approach for not only this manuscript, but for future investigation in these cell lines. We have also maintained this subtype level analysis in newly added analysis of 50+ additional human GSC lines, shown in Fig. 2 and Supplementary Fig. 2.

5) The analysis in Figure 7 attempts to show a direct effect between TrkB isoforms and gene expression via RNA-seq after infection in a neural stem cell line. However, the analysis seems very weak overall and the conclusions are blown way out of proportion for a such a limited study in a single cell line without meaningful biological replicates. Also it is unclear why NTRK2 transcript abundance appears to be increased only after infection with TrkB.T1 and not TrkB.FL. What's the deal with this? Doesn't this call into question the validity of the entire figure?

We apologize for the visually misleading color scheme in our heatmap. We have included a supplementary datasheet showing NTRK2 expression in these cells (NTRK2 is significantly upregulated in both TrkB.FL and TrkB.T1 transduced cells lines compared to GFP control) and adjusted the colors of the heatmap so that the shifts from downregulated to upregulated are not as visually drastic, which allows for a more appropriate and linear interpretation of the data that was perceptually confounded with our initial blue/red coloring scheme. Additionally, as defined in our methods, these cells were screened using a puromycin selection so would not be viable without lentiviral transduction of appropriate vectors (pLJM1-GFP, pLJM1-TrkB.FL, or pLJM1-TrkB.T1).

Additionally, we have now included a western blot from NSC lysates highlighting that TrkB.FL is increased in pLJM1-TrkB.FL transduced lines only (and not in pLJM1-TrkB.T1 lines), while TrkB.T1 levels are increased in pLJM1-TrkB.T1 transduced lines (and not in pLJM1-TrkB.FL lines), relative to each other and pLJM1-GFP transduced NSCs.

Reviewer #3 (Remarks to the Author):

In the manuscript entitled A kinase-deficient NTRK2 splice variant predominates in glioma and amplifies several oncogenic signaling pathways, Dr. Pattwell and colleagues evaluate the role of the TRKB splicing variant TrkB.T1 in human and mouse brain tumors with focus on glioblastoma (GBM) and Lower grade Glioma (LGG).

Given the important role of neurotrophins and their receptors (TRKA/TRKAB/and TRKC) in the development of the nervous system and the fact that fusions involving genes encoding the TRK receptors have been found as recurrent genetic drivers in brain tumors, the manuscript focuses on a topic of extreme scientific interest and clinical relevance.

However, I have several major concerns regarding the approach taken as well as the experimental validation of the authors' observations that I personally do not think always support their conclusions.

Major points

1) As an initial broad analysis, the authors analyze the expression levels of the whole gene NTRK2 across normal brain, LGG and GMB using two distinct datasets [GTEx and TCGA (2 studies)]. The expression of the whole gene was unchanged across normal brain, LGG and GBM. Nevertheless, the authors speculate that a differential expression of the different splicing variants of this gene may exist and compare two of them, the TrkB-FL and the kinase-deficient TrkB.T1. This is a biased choice that is just justified by the fact that these two isoforms are the most studied. To give a stronger rationale on why the authors further focus on the TrkB.T1 isoform, the analysis should have included at least all the splicing variants of TrkB (if not all the variants of all NTRK genes) (PMID: 30333516). If TrkB.T1 confirms to be the most highly differentially expressed isoform, the rationale to focus on this variant would be much stronger than it is right now.

We thank the Reviewer for this suggestion. While we did explore all TRK variants prior to starting these studies, we neglected to include these data in our manuscript and agree that it makes the rationale for focusing on this variant much stronger. We have provided these data in an additional supplementary datasheet (Supplementary Datasheet 2) which shows the expression of all NTRK1, NTRK2, NTRK3 isoforms.

Additionally, to further expand beyond TCGA transcript data to further confirm the importance of this particular variant, we performed additional screens and experiments looking at TRK expression in a variety of mouse tumorspheres and a wide range of human glioblastoma stem cells (GSCs) (50+ lines in total) and have included this data in our revised manuscript (Fig. 2, Supplementary Datasheets 2 and 3, Supplementary Fig. 2,

Supplementary Fig. 7). All mouse tumorspheres (with the exception of one NTRK1 fusion line for which TrkB.T1 expression was the second highest expressed after NTRK1 (TrkA) kinase) and all 50 lines analyzed show TrkB.T1 to be the most highly expressed variant.

2) The authors use WB analysis to show that TrkB.T1 expression at protein level is higher in glioma when compared to normal human cortex (fig. 2C). They discriminate between the FL and the T1 forms based on the size. There is overlap between the two band in sample 2 (glioma), thus rendering very hard to understand the differences in levels compared to the normal. Moreover, this is just 1 sample vs 1 control. This WB is not informative and should be removed.

We have removed this western blot, as requested by Reviewer #3. Additionally, we have included the additional data described in point #1 for 50+ human GSC lines and mouse tumorspheres and present this as transcript data, western blot, and qPCR (Fig. 2, Supplementary Datasheets 2 and 3, Supplementary Fig. 2, Supplementary Fig. 7).

In regards to this important distinction between TrkB.T1 and TrkB.FL, as requested by Reviewer # 2, we have also now included IHC for the TrkB.FL using an antibody designed against the kinase of this receptor (specifically amino acid ~810 of TrkB) and it very clearly shows a distinct pattern from that of TrkB.T1. Unlike the pattern observed with TrkB.T1, TrkB.FL is present in the TrkB.T1^{-/-} cortex, present in WT cortex, but absent in tumors. These collective images are found in Supplementary Fig. 5 and Supplementary Fig. 6.

3) The authors realize the need of a better tool to discriminate the different isoforms and design a T1-specific antibody that recognize the unique C-term of this variant. They validate its specificity in different ways using tissues collected from mice with different TrkB-T1 background. The validation here is crucial but Supplementary figure 3 is very blurry. Also, I think that now the authors should present the WB (Fig 2C), with the normal antibody as well the TrkB.T1 specific. In this case just 1 band in both samples should show up with the T1-specific and maybe the differential abundance among the normal and the tumor can be better appreciated.

We apologize that Supplementary Fig. 3 became blurry after uploading and have addressed this in the resolution of our revised files by attempting to maintain file size during the upload process.

4) The authors stained multiple glioma models with different background (EGFR, PDGF, PTEN null, NF1 null) with the T1 antibody and found strong, similar positivity in all samples. However, for all the following experiments, they focused on a PDGF-driven model? Which is the rationale? EGFR signaling is often altered in glioma and, as shown in the results presented in Figure 3D (GO analysis), EGFR signaling is a hit as a differentially expressed pathway between tumor and normal. This is again a biased choice that should be avoided or, alternatively, other hits should be studied further for comparison. For example, transduced (with FL and T1) 3T3 and NSCc should be tested

following EGF and BDNF (as a control) stimulation. Also, the authors suggest a cross-talk between T1 and RTKs (overall PDGF). CO-IPs should be performed to test whether they bind.

We thank the Reviewer for raising these points. As PDGFB is widely and reliably used with the RCAS-tv/a system for glioma modeling, in our lab and others, we focused on this for the majority of *in vivo* and *in vitro* studies. We wanted to see if there were any tumor types that did not exhibit strong TrkB.T1 staining or if this pattern may be inclusive of various tumor types. Supplementary Fig. 5 was intended to be a survey of glioma models to see if other tumor types – despite strain or RCAS virus – would also exhibit the patterns we see in PDGFB driven tumors and in fact, strong, diffuse TrkB.T1 staining was observed in all models.

We tested BDNF (at 100ng/ml) to ensure this effect is BDNF independent and did not see any changes in pAkt, which was not surprising as 3T3 cells do not express endogenous TrkB.FL. We do feel confident, based on our *in vitro* data, that this PDGF-driven effect observed in our mouse model is not driven by TrkB.T1-TrkB.FL heterodimers because there is no TrkB.FL in these cells and as such, TrkB.FL cannot respond to this ligand if it is not present. We have emphasized this in our revised text, through a newly added reference (Fryer, R.H., Kaplan, D.R. & Kromer, L.F. *Exp Neurol*, 148: 616-27 (1997)) and as can be shown in Fig. 5b. Similarly, 3T3 cells do not express endogenous EGFR and would not be responsive to EGF ligand (Hudson et al., *Cancer Res*, 74(21):6271-6279 (2015)), so PDGFB was preferable to be able to perform parallel *in vitro* experiments in these cells.

5) Data presented in Figure 6 on the 448T and 559T cells are not supportive of the authors' conclusions. These cell lines were chosen for their differential expression of T1 and compared at the level of transcriptome. The fact that two different cell lines have differential gene expression is absolutely normal and the authors can not conclude that these differences are due to the different expression of T1. If the authors want to claim this, they should, for example, knock-down T1 (maybe targeting the unique C-term sequences) in the high T1-expressor 448T, check the KD with their new antibody and then run RNA seq on this pair. The same can be done using the other cell line following T1 overexpression. If these 2 experiments are not performed the whole section is inconclusive and needs to be removed.

We thank the Reviewer for this suggestion. In accordance with Reviewer #1's suggestion, we have included analysis of two additional human GSC lines for DGCA (differential gene correlation analysis). To ensure we were not biasing our data, we chose GSC human cell lines derived from distinct tumor subtypes, in this case, with a mesenchymal phenotype (G14 and G179), to compare to our initial proneural/classical lines (448T and 559T). We have included this new data in the revised manuscript, comparing the different GSC lines to each other, and also the high TrkB.T1 expressing lines to the low TrkB.T1 expressing lines, and believe this addition of not only adding two additional lines, but also two distinct

types of GSCs (mesenchymal and proneural) strengthens our GSC approach for not only this manuscript, but for future investigation in these cell lines. We have also maintained this subtype level analysis in newly added analysis of 50+ additional human GSC lines, shown in Fig. 2 and Supplementary Fig. 2.

6) Following the final transcriptomic analysis on the NSCs transduced with FL and T1, the authors conclude that the PDGF/PI3K/AKT/mTOR axis is activated preferentially in the presence of T1. No validation is presented. Moreover, if this is correct, these cells should respond better to PDGF/AKT/PI3K/mTOR inhibitors when compared to the FL transduced cells. These experiments as proliferation assays and biochemical WB-based assays should be performed.

We thank the Reviewer for this suggestion and have treated human neural stem cells (NSCs) transduced with either TrkB.FL or TrkB.T1 with varying concentrations of PI3K inhibitor (LY294002) or mTOR inhibitor (rapamycin) and tested for proliferation differences and viability via CellTiter-Glo assay which determines the number of viable cells in a culture based on quantification of ATP, an indicator of metabolically active cells. As these NSCs are stem-like by nature, both drugs worked in a dose dependent manner to reduce viability and TrkB.T1-transduced NSCs were marginally, but statistically significantly, more sensitive to LY294002, which is in line with data throughout the manuscript suggesting a role for TrkB.T1 in modulating PI3K signaling.

This is shown in our revised manuscript:

“Treatment of NSCs transduced with TrkB.FL and TrkB.T1 showed that both lines were sensitive to phosphoinositide-3 kinase (PI3K) and mTOR inhibitors (LY294002 and rapamycin, respectively) in a dose dependent manner and that TrkB.T1 NSCs were marginally, but statistically significantly, more sensitive to LY294002 than TrkB.FL NSCs at both 48 and 72 hours (Supplementary Fig. 8d, 8e). Combined with RNASeq data suggesting a role for TrkB.T1 in PI3K signaling and enhancement of downstream PDGFR pathways *in vitro* (Fig. 5, Fig. 7, Supplementary Fig. 8, Supplementary Datasheet 6), increased sensitivity to LY294002 in the TrkB.T1 NSCs suggests that TrkB.T1 may selectively activate PI3K pathways.”

We also explored our RNASeq data for a proliferation index (as previously described in newly added references Ramaker et al. (2017) *Oncotarget* (8)24: 38668-38681 and Venet et al. (2011) *PLoS Comput Biol*(7)10: e1002240). As these NSCs are stem-like by nature, proliferation indices are high overall due to the cell type and culture conditions required for stemness, however, TrkB.T1 transduced NSCs do show higher expression of genes in this proliferation index compared to TrkB.FL as shown in Supplementary Fig. 9.

7) Figure 7C the expression of TRKB in the T1 transduced cell is much higher than in

cells transduced with TrkB FL. The WB showing the transduced cell line is required since these cells are compared

We apologize for the visually misleading color scheme in our heatmap. We have included a supplementary datasheet showing NTRK2 expression in these cells (NTRK2 is significantly upregulated in both TrkB.FL and TrkB.T1 transduced cells lines compared to GFP control) and adjusted the colors of the heatmap to not be as visually drastic and allow for an appropriate linear interpretation of the data that the blue/red scheme confounded. Additionally, as defined in our methods, these cells were screened using a puromycin selection so would not be viable without lentiviral transduction of appropriate vectors (GFP, pLJM1-TrkB.FL, or pLJM1-TrkB.T1).

Additionally, we have now included a western blot from NSC lysates highlighting that TrkB.FL is increased in pLJM1-TrkB.FL transduced lines only (and not in pLJM1-TrkB.T1 lines), while TrkB.T1 levels are increased in pLJM1-TrkB.T1 transduced lines (and not in pLJM1-TrkB.FL lines), relative to each other and pLJM1-GFP transduced NSCs.

Minor points:

1) On page 3, when “several, lesser known, alternative spliced variants” are mentioned, the statement should be referenced (For example PMID: 30333516)

Although this reference tends to focus more on gene fusions than alternative splicing events, we have included it, along with other references, when referencing the other NTRK variants, in our revised manuscript.

2) I am not sure Fig 1b should be a main one. It is already clear from the 1A that the rest of the samples (once cerebellum and spinal cord data are removed) cluster together. I would move to the Suppl.

Due to a limitation in the number of supplementary figures (one supplementary figure per main figure), we have retained this in the Main Figure as the inclusion of additional data requested here for this revision has restricted our ability to add more to data to our Supplementary Figures and files.

3) Please indicate statistic for the survival curve presented in suppl Fig 1. Also, I think this is an interesting point that the authors may want to highlight as it may have strong clinical implications (T1 could be a biomarker of prognosis tested with the T1-specific antibody)

We have included statistics on this survival curve, as well as separated this into GBM and LGG, as well as highlighted this in the text with the overarching summary that high

expression of TrkB.FL does *not* lead to worse prognosis, as has been suspected in literature on NTRK fusions.

We also thank the Reviewer for the suggestion of potentially looking at TrkB.T1 as a biomarker and are currently developing methodology and IRB applications for detecting TrkB.T1 in glioma patients through various methods using our custom antibody.

4) I am not sure Suppl Fig 4C have the images in the same magnification when compared with panel A and B. Please change for consistency.

We have ensured that the magnification is listed appropriately in the Figure legends. These panels that remain with different magnifications do so as they were chosen by our neuropathologist to best show particular features (i.e punctate distribution in Supplementary Fig. 4b at 1000X vs blood vessels at much lower magnification 10-20X in Supplementary Fig. 4c.). Due to a limitation of supplementary figures, we had to include these items on the same page.

5) In the WB conducted on the transduced 3T3 the authors should show pPDGF, total PDGF, pERK and ERK.

We thank the Reviewer for this suggestion and have included WB for pPDGFR β (Y1021) and total PDGFR β as this seemed most relevant to the overall text of the main manuscript. This new data suggests that TrkB.T1 is capable of enhancing levels of PDGFR β in 3T3-tva cells and these are maintained even after PDGF-BB ligand administration.

Additionally, we transitioned our pERK western protocol from using one pERK (Cell Signaling #9102 at 1:1000) to an enhanced pERK antibody (Cell Signaling phospho-p44/42 MAPK (ERK1/2) (Thr202/Tyr204) (D13.14.4E) XP[®] #4370 at 1:2000) and have included this new data in the revised manuscript, in Fig. 5d.

6) Figure 4 the authors show no staining with T1 antibody in the normal. It would be nice to see the staining for the FL.

We apologize for any oversight and have clarified that we do show TrkB.T1 in normal brain in Fig. 4 and corresponding manuscript text. This pattern of TrkB.T1 in normal brain is punctate and less intense/diffuse than in gliomas. We have added additional immunohistochemistry for TrkB.FL, using an antibody designed against ~aa 810 of the TrkB kinase (abcam #ab18987) which very clearly shows a distinct pattern from that of TrkB.T1 – the TrkB.FL is present in the TrkB.T1^{-/-} cortex, present in WT cortex, but absent in tumors. These collective images are found in Supplementary Fig. 5 and Supplementary Fig. 6.

7) In the discussion “the data shown here reveal a kinase-deficient isoform, T1, to be the predominant isoform in brain tumors compared to normal brain”. This is an overstatement since the authors just checked the expression of this variant.

As addressed in Main Point #1, we have included additional data showing that this variant is indeed the most highly expressed signal sequence containing isoform of all not only NTRK2, but all TRKs (NTRK1, NTRK2, and NTRK3).

8) Pag 15 “and future studies should explore if this IS a PDGF.....”

We have corrected this typographical error in our manuscript.

Reviewers' Comments:

Reviewer #1:

Remarks to the Author:

The authors did an exceptionally good job of addressing this reviewer's comments. I have further concerns or questions.

Reviewer #2:

Remarks to the Author:

I don't think this article represents a significant enough advance, nor does it provide the appropriate level of mechanistic evidence, to warrant publication in a Nature journal.

Figure 1 shows only some principal component analysis of previously published normal brain and glioma gene expression data that provides essentially no new meaningful scientific advances.

Figure 2 shows only that a TrkB splice isoform is differentially expressed in glioma compared to normal brain.

Figure 3 shows only correlative gene expression analysis from TrkB high and TrkB low glioma samples. This is merely correlative mumbo jumbo and is considered meaningless and misleading data that is not worthy of publication in a Nature journal. The authors do not show that the TrkB splice isoform actually causes any biologic effects in this figure (nor in Figures 1, 2, 4, or 6)

Figure 4 only shows that the authors have developed an antibody that specifically recognizes the TrkB splice isoform of interest. There is essentially no meaningful biologic insight provided by this figure.

Figure 5 is the only figure that provides (albeit somewhat limited) mechanistic insight into the role of the TrkB.T1 splice isoform in gliomas.

Figure 6 is again merely correlative gene expression analysis in TrkB.T1 high and low gliomas which again represents correlative mumbo jumbo. The authors do not perform any experiments that show any causation of TrkB.T1 on gene expression. Their repetitive use of correlative gene expression analysis is simply descriptive in nature, rather than mechanistic science that truly advances our understanding of the biology of TrkB splice isoforms. These descriptive correlative gene expression analyses are just not worthy of publication in a Nature journal.

Reviewer #3:

Remarks to the Author:

The manuscript of Dr. Pattwell and colleagues improved significantly after the review.

The authors addressed the majority of my concerns. It is overall remarkable the analysis the authors have done to show that TRKB-T1 is actually the most expressed variant in tumors vs normal tissues (across all TRK variants). Very impressive data that make the rationale of this study very solid.

Despite being significant, the difference in sensitivity to PI3K/mTOR inhibitors between control and TRKB-T1-expressing cells is quite disappointing. Hopefully, future studies will suggest more effective combinatorial therapies.

As a note, a could not see the figures 6 and 7 anywhere, including in the merged file so I can not review those figures.